



# Simulating stable carbon isotopes in the ocean component of the FAMOUS General Circulation Model with MOSES1 (XOAVI)

Jennifer E. Dentith[1], Ruza F. Ivanovic[1], Lauren J. Gregoire[1], Julia C. Tindall[1], and Laura F. Robinson[2]

[1]School of Earth and Environment, University of Leeds, Leeds, UK, LS2 9JT
[2]School of Earth Sciences, University of Bristol, Bristol, UK, BS8 1RJ

*Correspondence to*: Jennifer E. Dentith (eejed@leeds.ac.uk)

**Abstract.** Isotopic ratios are often utilised as proxies for ocean circulation and the marine carbon cycle. However, interpreting these records is non-trivial because they reflect a complex interplay between physical and biogeochemical processes. By directly simulating multiple isotopic tracer fields within numerical models, we can improve our understanding of the processes that control large-scale isotope distributions and interpolate the spatiotemporal gaps in both modern and palaeo datasets. We have added the stable isotope $^{13}$C to the ocean component of the FAMOUS coupled atmosphere-ocean General Circulation Model, which is a valuable tool for simulating complex feedbacks between different Earth System processes on decadal to multi-millennial timescales. We tested three different biological fractionation parameterisations to account for the uncertainty associated with equilibrium fractionation during photosynthesis and used sensitivity experiments to quantify the effects of fractionation during air-sea gas exchange and primary productivity on the simulated $\delta^{13}C_{DIC}$ distributions. Following a 10,000 year pre-industrial spin-up, we simulated the Suess effect (the isotopic imprint of anthropogenic fossil fuel burning) to assess the performance of the model in replicating modern observations. Our implementation captures the large-scale structure and range of $\delta^{13}C_{DIC}$ observations in the surface ocean, but the simulated values are too high at all depths, which we infer is due to biases in the biological pump. In the first instance, the new $^{13}$C tracer will therefore be useful for recalibrating both the physical and biogeochemical components of FAMOUS.



# 1 Introduction

Carbon isotopes are often used as proxies for ocean circulation and the marine carbon cycle. There are three naturally occurring carbon isotopes: the stable isotopes $^{12}C$ (98.9 %) and $^{13}C$ (1.1 %), and the radioactive isotope $^{14}C$ ($1.2 \times 10^{-10}$ %), which is also known as radiocarbon (Key, 2001). The relative proportions of $^{12}C$, $^{13}C$ and $^{14}C$ in a given oceanic pool (e.g.

dissolved inorganic carbon, DIC, or particulate organic carbon, POC) are controlled by ocean circulation and mixing, and mass dependent fractionation during biogeochemical processes such as air-sea gas exchange (Lynch-Stieglitz et al., 1995; Zhang et al., 1995), photosynthesis (e.g. Sackett et al., 1965; Rau et al., 1989; Hollander and McKenzie, 1991; Keller and Morel, 1999), and calcium carbonate formation (Emrich et al., 1970; Turner, 1982; Ziveri et al., 2003). This is typically reported in delta ($\delta$) notation, which is the heavy to light isotope ratio (R) of a sample relative to a standard in per mil (‰) units (($R_{sample}$ / $R_{standard}$

- 1) $\times$ 1000; Stuiver and Polach, 1977). In this study we focus on $\delta^{13}C$, which is primarily used to track individual water masses (Curry and Oppo, 2005), study past changes in the carbon cycle (e.g. de la Fuente et al., 2017), and investigate changes in ocean circulation on glacial-interglacial timescales (e.g. Spero and Lea, 2002; Campos et al., 2017). It has also been used to constrain air-sea gas exchange rates (Gruber and Keeling, 2001) and to estimate the uptake of anthropogenic carbon by the global oceans (Quay et al., 1992, 2003).

Oceanographic surveys conducted since the 1970s, such as the World Ocean Circulation Experiment (WOCE; Orsi and Whitworth III, 2005; Talley, 2007; Koltermann et al., 2011; Talley, 2013), and synthesis projects such as Carbon dioxide in the Atlantic Ocean (CARINA; Key et al., 2010), Pacific Ocean Interior Carbon (PACIFICA; Suzuki et al., 2013), and the Global Ocean Data Analysis Project (GLODAP; Key et al., 2004, 2015; Olsen et al., 2016), provide an indication of large-scale carbon isotope distributions in the modern oceans. The two main drawbacks of these surveys are that they include

relatively few measurements from the sub-surface ocean and that there were only a limited number of repeat measurements at fixed locations, which were often taken decades apart. These datasets are therefore insufficient for studying transient changes in carbon isotope distributions at sub-decadal resolution.

Geological archives such as corals (e.g. Guilderson et al., 2013) and sediment cores (e.g. Oliver et al., 2010) are used to extend the record further back in time. However, interpreting isotopic ratios in geological archives is non-trivial because

they result from a complex interplay between physical processes and biogeochemical processes, both in the water column itself and during biomineralisation, which can be difficult to disentangle.

By including carbon isotopes in climate models, we can fill in the spatiotemporal gaps in both modern and palaeo datasets, and improve our understanding of the processes that control their large-scale distributions (Tagliabue and Bopp, 2008; Schmittner et al., 2013; Menviel et al., 2017). The Ocean Carbon-Cycle Modelling Intercomparison Project (OCMIP) was

initiated in 1995 with the aim of evaluating the major differences between global ocean carbon cycle models and advancing our understanding of the ocean as a long-term $CO_2$ reservoir (Orr, 1999). Carbon isotopes are not routinely incorporated into climate models because of the computational expense associated with the long equilibration between the deep ocean and the



atmosphere (Bardin et al., 2014). However, since OCMIP produced a legacy of standard input fields (Orr, 1999; Orr et al., 2000, 2017), carbon isotopes have increasingly been implemented into models of varying complexities to validate physical and biogeochemical schemes, to investigate the spatiotemporal variability in isotope distributions, and to reconcile the interpretation of ocean proxy data. As outlined in Table 1, the community of $^{13}$C-enabled models currently includes:

HAMOCC3.1 (Hofmann et al., 2000), the GFDL modular ocean model (MOM; Murnane and Sarmiento, 2000), CLIMBER-2 (Brovkin et al., 2002), MoBidiC (Crucifix, 2005), PISCES (Tagliabue and Bopp, 2008), Bern3D+C (Tschumi et al., 2011), the UVic Earth System Model (ESM; Schmittner et al., 2013), iLOVECLIM (Bouttes et al., 2015), CESM (Jahn et al., 2015), and CSIRO Mk3L-COAL (Buchanan et al., 2019). Most of these are low resolution (3 to 5°), intermediate complexity models that are valuable tools for studying changes in ocean biogeochemistry on multi-millennial timescales. However, these models

do not provide sufficient complexity in the ocean circulation, vertical mixing and atmosphere-ocean interactions to study more abrupt (decadal-to-centennial) changes. The more complex models (e.g. PISCES and CESM) provide a more sophisticated representation of physical and biogeochemical processes because of increased spatial resolution and/or the inclusion of more carbon pools. However, these models are computationally expensive, for example, at the time of their study, a 6010 year spin-up simulation with CESM took over 7 months to run (Jahn et al., 2015). Without employing offline or accelerated spin-up

techniques (e.g. Lindsay, 2017), the higher complexity models are therefore less practical for running the long simulations required to fully spin-up the components of the Earth System that evolve on millennial timescales, such as deep ocean circulation (England, 1995) and ocean biogeochemical cycles (Falkowski et al., 2000; Key et al., 2004).

Here, we describe the implementation of $^{13}$C in the ocean component of the FAMOUS General Circulation Model (GCM). FAMOUS is well suited to studying ↕complex interactions between different components of the Earth System on

decadal to multi-millennial timescales, owing to its reduced spatial resolution and increased timestep relative to the latest generation of state-of-the-art GCMs (Sect. 2.1). We use sensitivity experiments to quantify the effects of isotopic fractionation during air-sea gas exchange and primary productivity on the simulated $\delta^{13}C_{DIC}$ distributions (Sect. 2.3.3 and Sect. 3.1), and test three different parameterisations for photosynthetic fractionation to account for the uncertainty associated with the relative influence of ambient conditions, physiological effects and transport mechanism on the fractionation of carbon isotopes during

photosynthetic $CO_2$ fixation (Sect. 2.2.2 and Sect. 3.3). We evaluate the overall performance of the model in simulating large-scale $\delta^{13}C_{DIC}$ distributions by comparing to modern observations (Sect. 3.2) and discuss the potential of the new $^{13}$C tracer as a tuning target for future recalibration work (Sect. 3.4).



## 2 Methods

### 2.1 Model description

FAMOUS is a coupled atmosphere-ocean GCM (Jones et al., 2005; Smith et al., 2008; Smith, 2012; Williams et al., 2013) based on HadCM3 (Gordon et al., 2000; Pope et al., 2000). Both are configurations of the UK Met Office Unified Model version 4.5 (Valdes et al., 2017). The quasi-hydrostatic primitive equation atmospheric model is 5° in latitude by 7.5° in longitude, with 11 vertical levels on a hybrid sigma-pressure coordinate system. The rigid-lid ocean model has a horizontal resolution of 2.5° × 3.75° and 20 unevenly spaced vertical levels, which are approximately 10 m thick in the near-surface ocean and 600 m thick in the deep ocean. The atmosphere and ocean operate on 1-h and 12-h timesteps, respectively, and are coupled once per model day. The model currently includes oxygen (Williams et al., 2014) and chlorofluorocarbons (Pope et al., 2000) as optional tracers. At the time of this study, FAMOUS is capable of simulating 400 to 500 model years per wallclock day on Tier 2 (regional) and Tier 3 (local) High Performance Computers at the University of Leeds, which is more than 5 times the run speed of HadCM3. This makes FAMOUS ideal for running long (multi-millennial) simulations (Smith and Gregory, 2012; Gregoire et al., 2012, 2015) or large (hundred-member) ensembles (Gregoire et al., 2011; Sagoo et al., 2013). Further technical documentation can be found in an ongoing special issue in Geoscientific Model Development (http://www.geosci-model-dev.net/special_issue15.html).

We added $^{13}$C as an optional passive tracer into the ocean component of FAMOUS, using the Met Office Surface Exchange Scheme (MOSES) version 1 (Cox et al., 1999) generation of the model to evaluate our scheme. Although a newer version of the land surface model exists, which includes the terrestrial carbon cycle and interactive vegetation (MOSES2.2; Essery et al., 2001, 2003; Williams et al., 2013; Valdes et al., 2017), problems have been identified with its representation of Meridional Overturning Circulation (MOC) in multi-millennial simulations with constant pre-industrial boundary conditions (Dentith et al., 2019). Specifically, FAMOUS-MOSES2.2 simulates a collapsed Atlantic MOC (AMOC) and a strong, deep Pacific MOC when the run length exceeds 6000 years, resulting in spurious ocean tracer distributions. However, our code is directly transferable between the different generations of the model, meaning that the isotope system can be extended into the terrestrial carbon cycle following additional tuning to improve the physical ocean circulation in FAMOUS-MOSES2.

### 2.1.1 Hadley Centre Ocean Carbon Cycle Model (HadOCC)

The marine carbon cycle in FAMOUS is modelled by HadOCC, a coupled physical-biogeochemical model that simulates air-sea gas exchange, the circulation of DIC, and the cycling of carbon by marine biota (Palmer, 1998; Palmer and Totterdell, 2001). The ecosystem model provides a simplified representation of the ocean biological system, with a single (nitrogenous) nutrient, a single class of phytoplankton, a single class of (non-migrating) zooplankton, and detritus. Changes in the size of these pools are calculated through a series of coupled differential equations that describe primary production, respiration, mortality, grazing, excretion, and the sinking and remineralisation of detritus. The system is nitrogen limited and



carbon flows are coupled to the nitrogen flows by stoichiometric ratios that are fixed for each pool of organic matter. In addition to the four biological components, HadOCC also explicitly simulates DIC and alkalinity. Modelled DIC concentrations depend upon phytoplankton growth and biological breakdown. Alkalinity is similarly affected by biological processes and is used to calculate the proportion of DIC that is in the form of $CO_2$ in the surface waters, and consequently the air-sea $CO_2$ flux. All six tracers are advected, diffused, and mixed across all levels, although phytoplankton and zooplankton concentrations are negligible below the euphotic zone (approximately the uppermost 100 m of the ocean). Detritus is the only biological tracer that is subject to sinking, which is parameterised at a constant rate of 10 m day$^{-1}$. However, there is no representation of sediments: any detrital material that reaches the ocean floor is therefore immediately refluxed back to the top layer of the ocean to conserve carbon and nitrogen. Calcium carbonate ($CaCO_3$) production is represented as an instantaneous redistribution of DIC and alkalinity below the lysocline, the depth of which is spatially and temporally constant (approximately 2500 m below sea level).

HadOCC accurately simulates low primary production in the sub-tropical gyres and high production in the regions with the greatest nutrient supply: the sub-polar North Pacific and North Atlantic Oceans, and around the Antarctic Convergence Zone (Figure 1). However, primary production is higher than observed in the eastern equatorial Pacific, which is attributed to excessive upwelling in the model (Palmer and Totterdell, 2001). Production is lower than observed northwards of 50° N in the Atlantic and Pacific basins because sea ice formation and melt do not affect salinity distributions. Consequently, stably stratified, low salinity layers of meltwater, which promote phytoplankton growth, are not represented in the model (Palmer and Totterdell, 2001). Furthermore, the simulated production in coastal regions is lower than observed. There are three main reasons for this: (1) HadOCC does not simulate riverine input of nutrients, which are a significant source of coastal nutrients; (2) most of the coastlines in FAMOUS are directly adjacent to ocean grid cells that are more than 1 km deep, which dilutes near-surface nutrient concentrations; and (3) upwelling is spread out over several grid points, which causes production to be more diffuse than observed (Palmer and Totterdell, 2001).

The level of representation of ecosystem processes in HadOCC is of intermediate complexity, making it computationally faster than more sophisticated ecosystem models that include additional POC species and/or multiple nutrients (e.g. PISCES). Errors in biogeochemical simulations are largely driven by biases in the physical ocean circulation (i.e. inaccuracies in the climate or ocean model to which the ecosystem model has been coupled; Doney, 1999; Doney et al., 2004; Najjar et al., 2007). Thus, simulating carbon isotopes in a more complex ecosystem model would not necessarily yield substantially better results.

## 2.2 Carbon isotope implementation

We added $^{13}$C to the four carbon pools in HadOCC: DIC, phytoplankton, zooplankton, and detritus (Figure 2). We assume that modelled DIC is $^{12}$C and carry $^{13}$C as a ratio (DI$^{13}$C/DI$^{12}$C), therefore virtual fluxes are not required to account for



the dilution or concentration effects of surface freshwater fluxes (Appendix A). We also use model units to minimise the error associated with carrying small numbers:

$$Model\ units = \frac{DI^{13}C}{DI^{12}C} \times \frac{100}{{}^{13}C/{}_{{}^{12}C_{std}}} \tag{1}$$

where ${}^{13}C/{}^{12}C_{std} = 1.12372 \times 10^{-2}$ (Craig, 1957). We account for isotopic fractionation during air-sea gas exchange (Sect. 2.2.1

and Appendix B) and photosynthesis (Sect. 2.2.2 and Appendix C). Observational estimates suggest that isotopic fractionation during $CaCO_3$ formation is between +3 ‰ and -2 ‰ (Ziveri et al., 2003), which is small compared to the other fractionation effects (Turner, 1982). Previous ${}^{13}C$ isotope implementation studies have therefore assumed either no isotopic fractionation during $CaCO_3$ production (Schmittner et al., 2013) or prescribed constant values, for example, +1 ‰ (Tagliabue and Bopp, 2008) or +2 ‰ (Jahn et al., 2015). We conducted sensitivity tests where fractionation during $CaCO_3$ formation was included

at constant rates of -2 ‰, 0 ‰ and +2 ‰, respectively. After 10,000 years, there was 0.001 ‰ difference in both the mean surface ocean $\delta^{13}C_{DIC}$ values and the surface standard deviations between all three simulations, and 0.02 ‰ difference between the three global volume-weighted integrals. Since these differences are small, we proceeded with the equivalent of no fractionation during $CaCO_3$ production ($\alpha_{CaCO3} = 1.0$).

### 2.2.1 Air-sea gas exchange

The air-sea gas flux of $DI^{12}C$ ($F$) is calculated as:

$$F = PV \times (C_{sat} - C_{surf}) \tag{2}$$

where $C_{sat}$ is the saturation concentration of atmospheric $CO_2$ (in mol m$^{-3}$), $C_{surf}$ is the surface aqueous concentration of $CO_2$ (in mol m$^{-3}$), and $PV$ is the piston velocity (in cm h$^{-1}$), which is calculated as:

$$PV = a \times u^2 \times (1 - a_{ice}) \times \left(\frac{Sc}{660}\right)^{-0.5} \tag{3}$$

where $a$ is a tuneable coefficient, $u$ is the wind speed (in m s$^{-1}$), $a_{ice}$ is the fractional ice cover and $Sc$ is the Schmidt number for $CO_2$, calculated as a function of sea surface temperature ($SST$, in °C):

$$Sc = 2073.1 - 125.62 \times SST + 3.6276 \times SST^2 - 0.043219 \times SST^3. \tag{4}$$

The air-sea gas flux of $DI^{13}C/DI^{12}C$ $\left(F_{\frac{13}{12}}\right)$ is calculated as:

$$F_{\frac{13}{12}} = \frac{1}{{}^{12}C} \times PV \times \left[ \alpha_k \times \alpha_{aq \leftarrow g} \times \left( C_{sat} \times \frac{{}^{13}A}{{}^{12}A} - \frac{C_{surf} \times \frac{{}^{13}C}{{}^{12}C}}{\alpha_{DIC \leftarrow g}} \right) - \left( \frac{{}^{13}C}{{}^{12}C} \times [C_{sat} - C_{surf}] \right) \right] \tag{5}$$

where ${}^{13}A/{}^{12}A$ and ${}^{13}C/{}^{12}C$ are the ${}^{13}C/{}^{12}C$ ratios of the atmosphere and DIC, respectively, $\alpha_k$ is the constant kinetic fractionation factor (0.99919), $\alpha_{aq \leftarrow g}$ is the temperature-dependent fractionation during gas dissolution:

$$\alpha_{aq \leftarrow g} = 0.9986 - (4.9 \times 10^{-6}) \times SST , \tag{6}$$





and $\alpha_{DIC\leftarrow g}$ is the temperature-dependent fractionation between aqueous $CO_2$ and DIC:

$$\alpha_{DIC\leftarrow g} = 1.01051 - (1.05 \times 10^{-4}) \times SST. \hspace{2cm} (7)$$

All three fractionation factors are based on the equations of Zhang et al. (1995). However, following Schmittner et al. (2013), we neglect the effect that the carbonate fraction ($fCO_3$) has on $\alpha_{DIC\leftarrow g}$ because this is much smaller (0.05 ‰) than the temperature effect (3 ‰). Currently, atmospheric $CO_2$ and $\delta^{13}C$ concentrations can either be held constant or prescribed from a file that contains a single global weighted-average value per year.

### 2.2.2 Photosynthesis

Isotopic fractionation during photosynthesis ($\alpha_{POC\leftarrow DIC}$, herein $\alpha_p$) is calculated as:

$$\alpha_p = \frac{\alpha_{aq\leftarrow g}}{\alpha_{DIC\leftarrow g}} \times \alpha_{POC\leftarrow aq} \hspace{2cm} (8)$$

where $\alpha_{POC\leftarrow aq}$ is the equilibrium fractionation factor between aqueous $CO_2$ and particulate organic carbon (POC).

Empirical relationships for the different biogeochemical fractionation effects ($\alpha_{aq\leftarrow g}$, $\alpha_{DIC\leftarrow g}$ and $\alpha_{POC\leftarrow aq}$) have been established from laboratory experiments, modern oceans and lakes, and the sedimentary record. However, there are still uncertainties associated with the parameterisation of $\alpha_{POC\leftarrow aq}$. Early studies investigated a potential temperature dependence of the carbon isotope composition of marine phytoplankton. For example, Sackett et al. (1965) proposed that photosynthetic fractionation is higher at lower temperatures (0.23 ‰ per °C) after observing that phytoplankton in the Drake Passage had more negative $\delta^{13}C$ values than those in the tropics. Wong and Sackett (1978) also recorded small temperature effects (-0.13 to +0.36 ‰ per °C) in 17 species of marine phytoplankton; however, the authors concluded that the 15 ‰ range observed in their samples was primarily related to different metabolic pathways within the organisms. Numerous studies have suggested that the fractionation of carbon isotopes during photosynthetic $CO_2$ fixation relates to aqueous $CO_2$ concentrations ($CO_2^*$) in the ambient environment (Popp et al., 1989; Rau et al., 1989; Jasper and Hayes, 1990; Hollander and McKenzie, 1991; Freeman and Hayes, 1992). However, these studies assumed that $CO_2$ only enters the phytoplankton by passive diffusion and neglected physiological effects, such as phytoplankton growth rate, cell size and geometry, and cell membrane permeability. Taking into consideration that physiological factors may modify, weaken, or eliminate the relationship between $CO_2^*$ and photosynthetic fractionation, Rau et al. (1996) proposed a model that accounted for the isotopic composition of the ambient aqueous $CO_2$, isotopic fractionation associated with diffusive transport into the cell, and isotopic fractionation associated with enzymatic, intracellular fixation. Laws et al. (1995) identified a linear relationship between phytoplankton growth rate, $CO_2^*$ and isotopic fractionation during photosynthesis, under the assumption that the growth rate is proportional to the net transport of $CO_2$ into the cell. A later study by Laws et al. (1997), which analysed the same species of marine diatom over a larger range of $CO_2^*$, revised this to a non-linear relationship. Burkhardt et al. (1999) and Keller and Morel (1999) additionally included active bicarbonate transport in their calculations, recognising that aqueous $CO_2$ is not the only substrate for photosynthetic fixation





and that processes other than diffusion can contribute to inorganic carbon acquisition. This has been a relatively inactive research area in the last 20 years, but there remains no single accepted model for fractionation during photosynthesis.

Consequently, previous carbon isotope implementation studies have used a number of different parameterisations for biological fractionation (Table 1), with the choice of scheme largely reflecting the complexity of the simulated biogeochemical

and ecosystem processes. It is difficult to compare the success of the different parameterisations used by individual modelling groups because inter-model differences in the simulated isotopic distributions predominantly relate to resolution, complexity, and biases in the physical ocean circulation and ocean biogeochemistry, as opposed to the choice of fractionation scheme. However, Hofmann et al. (2000) tested three different fractionation schemes within a single model. In their study, the oversimplified assumption of constant biological fractionation, taken from Maier-Reimer (1993), failed to reproduce the

observed latitudinal gradients in $\delta^{13}C_{POC}$. Calculating the fractionation as a function of $CO_2^*$, as per Popp et al. (1989), successfully replicated the interhemispheric asymmetry in $\delta^{13}C_{POC}$, but a growth rate dependent fractionation (e.g. Rau et al., 1996) was required to additionally capture the seasonal variations. Jahn et al. (2015) also demonstrated differences between three different fractionation schemes within a single model. In their study, the simple scheme of Rau et al. (1989) produced lower $\delta^{13}C_{DIC}$ values in the surface ocean and higher $\delta^{13}C_{DIC}$ values below 150 m compared to the more complex

parameterisations of Laws et al. (1995) and Keller and Morel (1999). The differences between the intermediate complexity formulation (Laws et al., 1995) and the most complex formulation (Keller and Morel, 1999) were small, and the Laws et al. (1995) equation was chosen as the default scheme.

To account for the uncertainty associated with biological fractionation in FAMOUS, we tested three different parameterisations for $\alpha_{POC\leftarrow aq}$. In the standard simulation (*std*), we calculated $\alpha_{POC\leftarrow aq}$ according to Popp et al. (1989):

$$\alpha_{POC\leftarrow aq} = -0.017\log\left(CO_2^*\right) + 1.0034 \qquad (9)$$

where $CO_2^*$ is the aqueous $CO_2$ concentration (in µmol L$^{-1}$).

Both of the alternative parameterisations calculated $\alpha_{POC\leftarrow aq}$ as a function of the phytoplankton specific growth rate (µ) and $CO_2^*$, representing an increase in complexity relative to the standard scheme. The first was a linear relationship derived from the experimental results of Laws et al. (1995):

$$\alpha_{POC\leftarrow aq} = \frac{-15}{\left(\mu/CO_2^*\right) - 15.371}. \qquad (10)$$

The second was a non-linear relationship derived from the experimental results of Laws et al. (1997):

$$\alpha_{POC\leftarrow aq} = \frac{1 + \left(\mu/0.225CO_2^*\right)}{1.0268 + 1.0055\left(\mu/0.225CO_2^*\right)}. \qquad (11)$$

Because HadOCC is a relatively simple ecological model, with only a single representation of phytoplankton, we did not test more complex schemes, such as those that use phytoplankton type-specific cell parameters (e.g. Burkhardt et al., 1999;

Keller and Morel, 1999).



### 2.2.3 Advection

The default advection scheme in FAMOUS is Quadratic Upstream Interpolation for Convective Kinematics (QUICK) with flux limiter (Leonard et al., 1993). This scheme is used to compute the transport of tracers such as temperature, salinity, nutrients, and DIC throughout the ocean. For consistency, we use the same advection scheme to calculate $^{13}C$ concentrations in the ocean interior. For greater numerical stability, $\delta^{13}C_{DIC}$ is fixed at 0 ‰ in the Hudson Bay and Baltic Sea. With the model's standard preindustrial land-sea mask, these inland bodies of water are isolated from the global oceans, therefore their isotope concentrations will not affect large-scale tracer distributions.

### 2.3 Simulations

### 2.3.1 Spin-up simulation

Carbon isotope simulations must be spun up over multiple millennia (5000 to 15,000 years; Orr et al., 2000) to reach steady state because of the long timescale of deep ocean ventilation (Bardin et al., 2014). We therefore ran our spin-up simulation for 10,000 years with constant pre-industrial boundary conditions, where $\delta^{13}C_{atm}$ was fixed at -6.5 ‰ (Francey et al., 1999) and $\delta^{13}C_{ocn}$ was initialised at a globally uniform value of 0 ‰. The global volume-weighted integral of $\delta^{13}C_{DIC}$ started to stabilise after 7000 years, and at the end of the spin-up simulation, the drift was less than 0.001 ‰ yr$^{-1}$ (Figure S1).

### 2.3.2 Historical simulation

A transient simulation for the period 1765 to 2000 CE was initialised from the end of the spin-up simulation to generate model output that is directly comparable to modern observations (Figure 3). Atmospheric $CO_2$ concentrations were prescribed from the OCMIP-2 files (Orr et al., 2000) and $\delta^{13}C_{atm}$ was prescribed from the Law Dome and South Pole ice core records (Rubino et al., 2013). The decrease in $\delta^{13}C_{atm}$ from -6.5 ‰ in 1750 to approximately -8.0 ‰ in 2000 is due to the Suess effect. First observed in tree ring records of atmospheric composition, the Suess effect refers to the dilution of $^{13}C$ in any carbon pool due to fossil fuel burning (Suess, 1955; Keeling, 1979). Fossil fuels formed millions of years ago from organic matter, which is relatively $^{13}C$-depleted due to isotopic fractionation during photosynthesis. Their isotopic signature is therefore approximately 20 ‰ lower than that of the ambient atmosphere (Andres et al., 1994, 1996). To act as a control, the spin-up simulation was continued for an additional 235 years with constant $CO_2$ and $\delta^{13}C_{atm}$.

### 2.3.3 Sensitivity experiments

Five further simulations were conducted to quantify the effects of fractionation during air-sea gas exchange and primary productivity on the simulated $\delta^{13}C_{DIC}$ distributions. All five simulations were run for 10,000 years with constant pre-industrial boundary conditions. In each of the simulations, $\delta^{13}C_{atm}$ was fixed at -6.5 ‰ and $\delta^{13}C_{ocn}$ was initialised at 0 ‰. At





the end of each of the spin-up simulations, the global volume-weighted $\delta^{13}C_{DIC}$ integral was drifting by less than 0.001 ‰ yr$^{-1}$.

Three of the simulations were designed to quantify the effects of the individual processes outlined in Sect. 2.2 (Table 2). In the *ki-fract-only* simulation, $\alpha_{aq\leftarrow g}$, $\alpha_{DIC\leftarrow g}$, and $\alpha_p$ were all set to 1, therefore only kinetic fractionation effects were

calculated. In the *no-asgx-fract* simulation, $\alpha_k$, $\alpha_{aq\leftarrow g}$, and $\alpha_{DIC\leftarrow g}$ were all set to 1 to eliminate the effect of fractionation during air–sea gas exchange. Fractionation during photosynthesis continued to be calculated using the *std* biological fractionation scheme, as per Eq. (8 − 9). In the *no-bio-fract* simulation, $\alpha_p$ was set to 1 to remove the effect of fractionation during photosynthesis, but fractionation during air-sea gas exchange continued to be calculated as per Eq. (5 − 7).

The other two simulations were designed to assess the sensitivity of the simulated $\delta^{13}C_{DIC}$ distributions to the choice

of biological fractionation scheme (Sect. 2.2.2). In the *L95* simulation, $\alpha_{POC\leftarrow aq}$ was calculated using Eq. (10), whilst in the *L97* simulation, $\alpha_{POC\leftarrow aq}$ was calculated using Eq. (11). As with the *std* simulation, we initialised a 235 year transient simulation (with the $^{13}$C-Suess effect) from the end of both of these spin-ups to allow the output from all three photosynthetic fractionation schemes to be compared directly to observations.

## 3 Results and discussion

### 3.1 Validating the isotope scheme

Isolating the different fractionation effects allows us to assess the relative contribution of air-sea gas exchange and biology to the simulated $\delta^{13}C_{DIC}$ distributions, and validate that the new isotope scheme is responding to physical and biogeochemical processes as expected. If there is no fractionation during either air-sea gas exchange or photosynthesis, the ocean equilibrates at a uniform value of -6.5 ‰, in line with the atmosphere. Kinetic fractionation has only a minor effect on

surface ocean $\delta^{13}C_{DIC}$ distributions, with simulated $\delta^{13}C_{DIC}$ values in the *ki-fract-only* simulation ranging between -6.57 ‰ in the Labrador Sea and -6.42 ‰ in the eastern equatorial Pacific (Figure 4a). This represents a -0.07 ‰ to +0.08 ‰ shift relative to no isotopic fractionation. Specifically, there is $^{13}$C depletion (low $\delta^{13}C_{DIC}$) in areas of net $CO_2$ invasion, such as the extra-tropics and high latitudes, and $^{13}$C enrichment (high $\delta^{13}C_{DIC}$) in the equatorial upwelling zones and the deep water formation regions where $CO_2$ is being outgassed. Kinetic fractionation has a negligible effect on the $\delta^{13}C_{DIC}$ depth profile, with globally

averaged $\delta^{13}C_{DIC}$ values of -6.4955 ‰ in the surface ocean and -6.5011 ‰ in the abyssal ocean (Figure 5).

When both the equilibrium and kinetic fractionation effects are included during air-sea gas exchange (*no-bio-fract*), the large-scale $\delta^{13}C_{DIC}$ distributions are closely related to the SST patterns because of the temperature dependence of $\alpha_{aq\leftarrow g}$ and $\alpha_{DIC\leftarrow g}$ (Figure 4b). Relatively high $\delta^{13}C_{DIC}$ values (> +2.5 ‰) are simulated in the Southern Ocean due to the combined effect of $CO_2$ outgassing and low SSTs, both of which cause $^{13}$C enrichment. The $\delta^{13}C_{DIC}$ values in the Arctic Ocean are comparably

low because the model has more extensive sea ice in the Northern Hemisphere than in the Southern Hemisphere, which inhibits





air-sea gas exchange. The highest values (+3.00 ‰) are simulated in the eastern equatorial Pacific where there are high rates of net $CO_2$ outgassing and Antarctic Bottom Water (AABW), which has a high $\delta^{13}C_{DIC}$ signature, is upwelled. Low $\delta^{13}C_{DIC}$ values are simulated in the Indian Ocean, with the lowest values (+1.1 ‰) in South East Asia, because the sea surface is warmer than at the equivalent latitudes in the Atlantic and Pacific Oceans. The globally averaged $\delta^{13}C_{DIC}$ values in this

simulation range between +2.03 ‰ in the surface ocean and +2.16 ‰ in the deep ocean, with a minimum value of +2.00 ‰ at a depth of approximately 200 m (Figure 5). Below approximately 1500 m, the globally averaged $\delta^{13}C_{DIC}$ is near constant with depth, matching the simulated temperature profile.

In the *no-asgx-fract* simulation, $\delta^{13}C_{DIC}$ values in the surface ocean range between -7.65 ‰ in the eastern equatorial Pacific and -3.89 ‰ in the eastern equatorial Atlantic (Figure 4c), representing a shift of -1.15 ‰ to +2.61 ‰ relative to no

isotopic fractionation. The asymmetry between these two upwelling zones occurs because the waters that are being upwelled from the deep Pacific Ocean are approximately 600 years older than the equivalent waters in the Atlantic Ocean. They therefore contain a higher percentage of remineralised organic matter, which is enriched in $^{12}C$. Relatively low $\delta^{13}C_{DIC}$ values are also simulated in the Southern Ocean and northeast North Atlantic Ocean where older water is mixed upwards from the abyssal ocean to the surface ocean at sites of deep water formation. The globally averaged $\delta^{13}C_{DIC}$ values in this simulation range

between -5.85 ‰ in the productive surface ocean and -7.56 ‰ in the abyssal ocean, with a minimum value of -7.86 ‰ at a depth of approximately 1000 m, which corresponds to the depth of maximum remineralisation in the model (Figure 5). The values change from greater than -6.5 ‰ (enriched in $^{13}C$ relative to no fractionation) to less than -6.5 ‰ (depleted in $^{13}C$ relative to no fractionation) at a depth of approximately 100 m, which corresponds to the photic zone.

The spatial patterns in the *std* simulation and the *no-asgx-fract* simulation are closely matched, both in the surface

ocean (Figure 6) and at depth (Figure 5), demonstrating the importance of biology to the large-scale $\delta^{13}C_{DIC}$ distributions. However, in the surface layer, air-sea gas exchange partly compensates for the biological effects in the Southern Ocean, the Northern Hemisphere deep water formation region, and the equatorial upwelling zones, as inferred from the peak surface zonal mean $\delta^{13}C_{DIC}$ values at 60° S, 55° N and 0° in the *no-bio-fract* simulation, which correspond with reduced amplitude troughs in the *std* simulation relative to the *no-asgx-fract* simulation. Similar results pertaining to the relative influence of air-sea gas

exchange and biology were presented by Schmittner et al. (2013), who concluded that air-sea gas exchange and temperature-dependent fractionation reduce the spatial $\delta^{13}C_{DIC}$ gradients that are created by biology. Earlier work by Murnane and Sarmiento (2000) and Schmittner et al. (2013) also supports the notion that biology is the dominant factor controlling $\delta^{13}C_{DIC}$ distributions in the interior ocean. Overall, the sensitivity experiments demonstrate that the new carbon isotope scheme is accurately responding to physical and biogeochemical processes in the model, such as temperature, air-sea gas exchange, and

the biological pump.



### 3.2 Comparison to observations

To assess the model performance in representing modern large-scale $^{13}$C distributions, we compare the simulated mean $\delta^{13}C_{DIC}$ values for the 1990s with observations from GLODAP version 2 (v2; Key et al., 2015; Olsen et al., 2016) and the gridded global ocean climatology of Eide et al. (2017). The $\delta^{13}C_{DIC}$ values in the *std* simulation are, on average, 0.97 ‰

higher than the GLODAPv2 observations in the surface ocean (Figure 7) and 0.64 ‰ higher globally, with root mean square error (RMSE) values of 1.03 ‰ and 0.91 ‰, respectively. However, the simulated range in the surface ocean (3.2 ‰) is in excellent agreement with the observed range (3.3 ‰). Specifically, the simulated surface $\delta^{13}C_{DIC}$ values are between +1.4 ‰ and +4.6 ‰, with a mean value of +2.6 ‰, whilst the observed surface $\delta^{13}C_{DIC}$ values range between -0.3 ‰ and +3.0 ‰, with a mean value of +1.5 ‰.

Re-examining the results of the sensitivity experiments allows us to ascertain the underlying causes of the model-data discrepancy. Schmittner et al. (2013; herein *S13*) conducted a similar set of simulations with the UVic ESM to elucidate the relative influence of biology and air-sea gas exchange on the distribution of oceanic $\delta^{13}C_{DIC}$ (see Table 1 in *S13*). Overall, there is good agreement between our *ki-fract-only* and *no-bio-fract* simulations and the equivalent simulations in *S13* (*ki-fract* and *no-bio*, respectively), both in the surface ocean and at depth. However, there is a clear difference between the results of our

*no-asgx-fract* simulation and the equivalent simulation in *S13* (*const-gasx*). Specifically, the surface ocean zonal mean $\delta^{13}C_{DIC}$ values in our *no-asgx-fract* simulation range between -6.6 ‰ at 60 °S and -5.5 ‰ in the sub-tropics, with a local minimum of -5.8 ‰ at the equator (Figure 6). For comparison, the surface ocean zonal mean values in *const-gasx* range between -8.0 ‰ in the Southern Ocean and -5.75 ‰ in the Southern Hemisphere sub-tropics, with a localised minimum of -6.25 ‰ at the equator (see Figure 4 in *S13*). Similarly, whilst the globally averaged deep ocean $\delta^{13}C_{DIC}$ values in our *no-asgx-fract* simulation have

a comparable range (2.01 ‰) to the deep ocean values in *const-gasx*, there is an offset of approximately 1 ‰, with *S13* simulating $\delta^{13}C_{DIC}$ values of -6.4 ‰ in the surface ocean, -8.4 ‰ in the deep ocean, and near constant values below 1000 m (see Figure 5 in *S13*). Overall, the $\delta^{13}C_{DIC}$ values in the standard simulation with the UVic ESM are in good agreement with observations, with a global linear regression $r^2$ value of 0.91 and a global RMSE of 0.33 ‰ (Schmittner et al., 2013; Buchanan et al., 2019). We therefore postulate that the offset in the simulated $\delta^{13}C_{DIC}$ values in FAMOUS relates to biases in the

biological carbon cycle.

Elucidating the exact cause of $\delta^{13}C_{DIC}$ model-data discrepancy is difficult. There are a number of fluxes in to and out of the DI$^{13}$C pool (Figure 2), each of which could have biases that are compounding or reducing the overall $\delta^{13}C_{DIC}$ bias. For example, if any of the rates of phytoplankton respiration, phytoplankton mortality or zooplankton mortality are too low, the input of $^{12}$C-enriched material back into the DIC pool would be insufficient. Similarly, if the model is not simulating enough

remineralisation, either as a direct consequence of the parameterised remineralisation rate or as a result of insufficient POC export, the input of $^{12}$C-enriched material back into the DIC pool would again be too low.





Primary producers preferentially take up $^{12}$C during photosynthesis, therefore higher than observed rates of net primary production in the photic zone would increase $\delta^{13}C_{DIC}$. However, if the $\delta^{13}C_{DIC}$ discrepancy in FAMOUS was a simple function of the biases in net primary production, $\delta^{13}C_{DIC}$ would be lower than observed in the subtropical gyres, the Indian Ocean, and the northern North Atlantic and North Pacific Oceans, and higher than observed in the equatorial upwelling zones
and the Southern Ocean (Figure 1). Thus, whilst the differences in net primary production could be contributing towards the $\delta^{13}C_{DIC}$ bias, particularly in the equatorial upwelling zones, they alone cannot explain the unidirectional offset.

Alternatively, the fractionation during photosynthesis could be too strong as a result of imbalances in the carbonate chemistry (Figure S2). The global mean alkalinity in FAMOUS is 81 µmol kg$^{-1}$ higher than observed and the mean alkalinity in the uppermost 50 m of the ocean is 107 µmol kg$^{-1}$ too high (Key et al., 2004; Sarmiento and Gruber, 2006). In addition, the
simulated global mean DIC concentration is 54 µmol kg$^{-1}$ higher than observed and the mean DIC concentration in the uppermost 50 m of the ocean is 96 µmol kg$^{-1}$ too high (Key et al., 2004; Sarmiento and Gruber, 2006). Furthermore, the mean ocean temperatures in FAMOUS are warmer than observed, both globally (2.2 °C) and in the uppermost 50 m of the ocean (1 °C; Sarmiento and Gruber, 2006; Locarnini et al., 2013). Increasing alkalinity increases $CO_2^*$, whilst increasing the temperature and DIC concentrations decreases $CO_2^*$. Hence, the overall effect of the carbonate chemistry biases in FAMOUS result in the
global mean $CO_2^*$ being 3.03 µmol L$^{-1}$ too low and the mean $CO_2^*$ in the uppermost 50 m of the ocean being 0.58 µmol L$^{-1}$ too high. In the photic zone, this translates to a simulated $\alpha_p$ of 0.97378 compared to an observed $\alpha_p$ of 0.97415 using the *std* fractionation parameterisation. Thus, we postulate that imbalances in the carbonate chemistry, and the consequent differences in $\alpha_p$, are contributing towards the $\delta^{13}C_{DIC}$ bias, but the overall effect is small.

The smallest model-data discrepancies in the surface layer are in the Southern Ocean and the northeast North Atlantic
Ocean where deep convection mixes $^{12}$C-enriched waters upwards (Figure 7). In contrast, in the equatorial upwelling zones, the effect of higher than observed primary productivity (increasing $\delta^{13}C_{DIC}$) outweighs the effect of vertical mixing (reducing $\delta^{13}C_{DIC}$), therefore the overall model-data biases are higher in these regions. Despite the global offset, the model correctly simulates lower $\delta^{13}C_{DIC}$ values in the Indian Ocean compared to the Atlantic and Pacific Oceans, because the Indian Ocean is relatively nutrient poor, both in the model and reality (Figure S3), which limits primary productivity (Figure 1). Similar to
previous $^{13}$C modelling studies (e.g. Hofmann et al., 2000; Tagliabue and Bopp, 2008; Schmittner et al., 2013), FAMOUS also accurately simulates the observed latitudinal gradient in mixed layer $\delta^{13}C_{POC}$, with relatively high values (≈ -20 ‰) in the low latitudes and relatively low values (≈ - 27 ‰) at high latitudes (Figure 8).

As observed, $\delta^{13}C_{DIC}$ decreases with depth in all basins due to the remineralisation of isotopically light organic matter (Figure 9). The maximum remineralisation depth in the model is approximately 1000 m, which is 200 to 500 m shallower than
observed. In the deep ocean, the highest $\delta^{13}C_{DIC}$ values are in the Atlantic basin, with intermediate values in the Indian basin, and the lowest values in the Pacific basin, where the waters are older and therefore contain more remineralised organic material (enriched in $^{12}$C). However, there are notable structural differences in the zonal means (Figure 10), which arise due to





inaccuracies in the physical ocean circulation in FAMOUS. Specifically, FAMOUS does not capture the observed structure in the Atlantic basin because, in this generation of the model, the AMOC is characterised by an over-deep North Atlantic Deep Water (NADW) cell and insufficient AABW formation (Dentith et al., 2019). FAMOUS also simulates weak (less than 3 Sv) ventilation to depths of 2000 m in the North Pacific Ocean (Dentith et al., 2019), which prevents the accumulation of old, $^{12}$C-enriched (low $\delta^{13}C_{DIC}$) waters at intermediate depths in the Northern Hemisphere. Instead, the oldest carbon in the model is in the eastern equatorial Pacific. In addition, the surface winds in the model are weaker than observed (Kalnay et al., 1996), resulting in a relatively shallow mixed layer. This promotes the excessive accumulation of high $\delta^{13}C_{DIC}$ values in the surface ocean, which is particularly notable in the Southern Hemisphere sub-tropical gyres. These physical model biases are also clearly visible in the zonal mean profiles of other tracers, such as nutrients (Figure S4) and DIC (Figure S5). The overall shape of the simulated depth profile reaffirms the notion that there are inaccuracies in both the physical and biogeochemical components of the model (Figure 9). Below approximately 1000 m, the simulated $\delta^{13}C_{DIC}$ values increase with depth in each ocean basin, whilst the observed basin averages are near constant with depth. The offset between the simulated and observed values is greatest in the deep Atlantic Ocean, where too much $^{13}$C-enriched water from the shallow ocean is being circulated into the abyssal ocean. However, the trend towards increasing $\delta^{13}C_{DIC}$ with depth could also be in-part explained by insufficient remineralisation in the model. This is supported by lower than observed nutrient concentrations in the deep ocean (Figure S4). HadOCC's global export ratio at 2000 m is within the observed range, but a lack of spatial variation means that the geographic distributions are partially incorrect (Palmer and Totterdell, 2001). Hence, we postulate that localised inaccuracies in the export ratio, together with deficiencies in the parameterisation of the remineralisation rate, are contributing towards the $\delta^{13}C_{DIC}$ offset. The basin-averaged $\delta^{13}C_{DIC}$ bias is smallest in the Pacific Ocean, where the waters are old and therefore have had more time to remineralise, thereby partially compensating for the biogeochemical biases. Indeed, the shape of the simulated and observed basin-averaged depth profiles are in good agreement below approximately 2000 m in the Pacific Ocean, despite the structural differences in the zonal mean.

As outlined in Sect. 3.1, our carbon isotope implementation is sensitive to physical and biogeochemical processes in the model. Thus, whilst biases in the overturning circulation and the biological pump are currently limiting the model's ability to accurately represent modern large-scale $^{13}$C distributions, the model-data agreement could be improved if the physical and ecological components of FAMOUS were recalibrated. This will be discussed further in Sect. 3.4.

### 3.3 Biological fractionation parameterisations

Given the uncertainty associated with biological fractionation (Sect. 2.2.2), we tested three different parameterisations for equilibrium fractionation during photosynthesis. For all three parameterisations, the total fractionation during photosynthesis is greatest in the high latitudes (where SSTs are relatively low and $CO_2^*$ is relatively high) and lowest in the equatorial regions (where SSTs are relatively high and $CO_2^*$ is relatively low; Figure 11). The *std* parameterisation produces





the largest range of $\alpha_p$ values (between approximately 0.97 and 0.98), whilst the *L95* parameterisation produces the smallest range (between approximately 0.964 and 0.970). The total fractionation during photosynthesis increases with the complexity of the parameterisation, with *L97* producing the largest overall effect (with a minimum $\alpha_p$ of 0.9635). For all three parameterisations, $\alpha_p$ decreases (i.e. the strength of fractionation increases) with depth in the photic zone, with the largest gradient produced by the *std* parameterisation (Figure S6).

The large-scale $\delta^{13}C_{DIC}$ patterns are very similar for all three photosynthetic fractionation schemes, but the parameterisations that take the phytoplankton growth rate in account simulate higher surface ocean $\delta^{13}C_{DIC}$ values everywhere except in the Southern Ocean, the Nordic Seas, and the eastern equatorial regions, where older $^{13}C$-depleted waters are mixed upwards from the abyssal ocean during deep water formation and upwelling (Figure 7). The differences are amplified when using the *L97* parameterisation (RMSE = 1.24 ‰, bias = 1.15 ‰), which specifies a non-linear relationship between μ and $CO_2^*$, compared to the *L95* parameterisation (RMSE = 1.21 ‰, bias = 1.13 ‰), which specifies a linear relationship (Figure S7). Conversely, the alterative parameterisations decrease $\delta^{13}C_{DIC}$ at depth compared to the *std* simulation, bringing the simulated values closer to the observations (Figure 9). Below approximately 500 m depth, the $\delta^{13}C_{DIC}$ values are consistently lower when using the *L97* parameterisation compared to the *L95* parameterisation. This is due to the preconditioning of $\delta^{13}C_{DIC}$ and $\delta^{13}C_{POC}$ as a result of fractionation during photosynthesis in the photic zone. In the *L95* and *L97* simulations, $\delta^{13}C_{POC}$ is lower than in the *std* simulation due to increased uptake of $^{12}C$ during primary production (lower $\alpha_p$). The latitudinal $\delta^{13}C_{POC}$ gradients in the mixed layer in these simulations are lower than observed, with zonal mean values ranging between approximately -30 ‰ at the equator and -33 ‰ at 60° N/S (Figure 8). When the POC is remineralised, a relatively low $\delta^{13}C$ signal is therefore being released back into the DIC pool, which causes the $\delta^{13}C_{DIC}$ in the deep ocean to be lower than in the *std* simulation. Thus, although the rates of biological exchange and overturning circulation are the same in all three simulations, the preconditioning of $\delta^{13}C_{DIC}$ and $\delta^{13}C_{POC}$ in the photic zone creates differences between the three simulations at depth. Whilst the global RMSE compared to the GLODAPv2 dataset is lower in the *L95* and *L97* simulations (0.86 ‰ and 0.87 ‰, respectively), it is still almost double the RMSE in other models (Buchanan et al., 2019). Overall, increasing the complexity of the fractionation scheme does not significantly improve the model-data agreement because of the aforementioned physical and biogeochemical biases.

**3.4 A new tuning target**

In this study, we have demonstrated that the new carbon isotope scheme in FAMOUS is sensitive to both physical and biogeochemical processes. The simulated $\delta^{13}C_{DIC}$ distributions therefore reflect known physical inaccuracies (such as over-deep NADW and weak convection in the sub-polar North Pacific Ocean) and have allowed us to identify previously





undisclosed biogeochemical biases (e.g. in the representation of remineralisation). The new tracer therefore offers excellent potential as a holistic tuning target for recalibrating FAMOUS in the future.

FAMOUS has previously been tuned both systematically (Jones et al., 2005; Gregoire et al., 2011; Williams et al., 2013) and manually (Smith et al., 2008). Most recently, Williams et al. (2013) tuned the 20 structural parameters in HadOCC

(coupled to FAMOUS-MOSES2.2) using an objective hypercube technique. Specifically, the parameter set included the C:N ratios for the different carbon pools, phytoplankton-specific parameters (e.g. maximum rate of photosynthesis), zooplankton-specific parameters (e.g. linear and quadratic zooplankton mortality rates), detritus-specific parameters (e.g. shallow and deep remineralisation rates), and carbonate-specific parameters (e.g. calcite export ratio). The main diagnostics used to evaluate the performance of the ensemble members were December-January-February and June-July-August surface air temperatures,

annual mean total precipitation rate, annual mean nitrate concentrations, and primary productivity. Crucially, this study only ran each perturbed parameter simulation for 200 years and neglected to evaluate the strength and structure of the AMOC. The optimal parameter set therefore had small but important imbalances in the surface climate, which caused the AMOC to collapse over longer (multi-millennial) timescales (Dentith et al., 2019).

HadOCC has not yet been tuned for the configuration of the model used in our study (FAMOUS-MOSES1).

Simultaneously recalibrating HadOCC and the physical ocean circulation in this generation of the model could therefore improve the simulated $\delta^{13}C_{DIC}$ distributions. We propose that the addition of $\delta^{13}C$ as a tuning target would improve the work of Williams et al. (2013) because it is an objective and straightforward way of assessing whether the balance between all of the ecological processes in the model is correct. We also suggest that implementing the radioactive isotope ($^{14}C$) into FAMOUS would be beneficial for future recalibration work (as well as subsequent scientific application of the isotope-enabled model)

because it is more sensitive to overturning circulation and air-sea gas exchange, and less sensitive to the biological pump, than $^{13}C$.

**Summary**

We have added the stable isotope $^{13}C$ to the ocean component of the FAMOUS GCM, using the MOSES1 generation of the model to validate our scheme. We account for fractionation during air-sea gas exchange and photosynthesis, and have

tested three different parameterisations for the latter. The model captures the range of observed $\delta^{13}C_{DIC}$ values in the surface ocean, but the simulated values are approximately 1 ‰ too high at all depths. The differences between the three fractionation schemes are relatively minor, but when fractionation during photosynthesis accounts for phytoplankton growth rates as opposed to just aqueous $CO_2$ concentrations the discrepancies between the model and observations are further increased in the surface ocean and reduced at depth. The sensitivity experiments suggest that the simulated values are too high because of

underlying biases in the biological carbon cycle, therefore retuning HadOCC could improve the model-data agreement. Biases in the large-scale ocean circulation also inhibit the model's ability to accurately simulate the large-scale distribution of tracers





in the deep ocean. Retuning the ocean circulation to improve the representation of the AMOC, in particular, would further reduce the model-data discrepancies. Thus, our results emphasise the utility of implementing carbon isotopes in GCMs; the simulated isotope distributions provide an additional measure against which the physical and biogeochemical model performance can be evaluated and offer an extra tuning metric for prospective development work. In the future, we intend to implement $^{14}$C following the same framework, before using the isotope-enabled model to study ocean circulation and the marine carbon cycle in both a modern and palaeo context, for example, at the Last Glacial Maximum (21,000 years ago) and during the last deglaciation (21,000 to 11,000 years ago).



**Code availability**

FAMOUS can be obtained from http://cms.ncas.ac.uk/wiki/UmFamous. The code detailing the advances described in this paper are available via the Research Data Leeds Repository (Dentith, 2019) under a Creative Commons Attribution 4.0 International (CC BY 4.0) license. These files are known as code modification ("mod") files and should be applied to the

5 original model code, which can be viewed online at http://cms.ncas.ac.uk/code_browsers/UM4.5/UMbrowser/index.html. All of the additional modification files that are required to run the simulations described in this manuscript are available in the Supplementary Material. These standard FAMOUS updates – some of which have been described by Smith et al. (2008), Smith (2012), and Valdes et al. (2017) – and the original model code are protected under UK Crown Copyright. The UM configuration ("basis") files for the simulations described in this paper are also available in the Supplementary Material.

**Table 3:** Overview of the simulations described in this study, as denoted by their unique five letter Met Office UM identifiers and the notation used within this manuscript.

| Identifier | Simulation | Duration |
|---|---|---|
| XOAVB | *std* spin-up | 0 to 10,000 years |
| XOAVI | *std* transient | 1765 to 2000 CE |
| XOGNC | *std* control | 1765 to 2000 CE |
| XOAVD | *ki-fract-only* | 0 to 10,000 years |
| XOAVE | *no-bio-fract* | 0 to 10,000 years |
| XOAVF | *no-asgx-fract* | 0 to 10,000 years |
| XOAVK | *L95* spin-up | 0 to 10,000 years |
| XOAVU | *L95* transient | 1765 to 2000 CE |
| XOAVL | *L97* spin-up | 0 to 10,000 years |
| XOAVW | *L97* transient | 1765 to 2000 CE |

**Data availability**

The data are available via the Research Data Leeds Repository (https://doi.org/10.5518/621).

15 **Author contributions**

RFI designed and supervised the project. JED wrote and implemented the code with input from JCT, LJG, and RFI. JED ran the simulations, analysed the results, and prepared the manuscript with input from all co-authors.



**Competing interests**

The authors declare that they have no conflict of interest.

**Acknowledgements**

JED was funded by the Natural Environment Research Council (NERC) SPHERES Doctoral Training Partnership (grant
number: NE/L002574/1). RFI acknowledges support from NERC grant NE/K008536/1. The contribution of JCT was
supported through the Centre for Environmental Modelling And Computation (CEMAC), University of Leeds. Numerical
climate model simulations made use of the N8 High Performance Computing (HPC) Centre of Excellence (N8 consortium and
EPSRC Grant #EP/K000225/1) and ARC2, part of the HPC facilities at the University of Leeds, UK. We thank Alexandra
Jahn (University of Colorado) for helpful discussions about implementing carbon isotopes in GCMs and Robin Smith
(University of Reading) for the supplying code used to cap the oceanic $CO_2$ flux.

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





**Tables**

**Table 1:** Overview of existing $^{13}$C-enabled models.

| Model | Horizontal resolution | Levels | Tracers | $\alpha_{POC\leftarrow aq}$ **parameterisation** |
|---|---|---|---|---|
| HAMOCC3.1 | $3.5° \times 3.5°$ | 15 | ALK[1], CaCO$_3$, DIC, $\delta^{13}$C$_{DIC}$, DOC[2], POC, $\delta$13C$_{POC}$, phytoplankton, zooplankton, PO$_4^{3-}$, H$_4$SiO$_4$, O$_2$ | Maier-Reimer (1993), Popp et al. (1989), Rau et al. (1996) |
| GFDL MOM | $4° \times 4°$ | 12 | ALK, DIC, DI$^{13}$C, DOC, DO$^{13}$C, PO$_4^{3-}$ | Freeman and Hayes (1992) |
| CLIMBER-2 | $2.5° \times 3$ zonally averaged basins | 20 | ALK, DIC, DI$^{13}$C, DI$^{14}$C, fast and slow DOC, DO$^{13}$C, DO$^{14}$C, PO$_4^{3-}$, O$_2$ | Rau et al. (1989) |
| MoBidiC | $5° \times 3$ zonally averaged basins | 19 | ALK, DIC, DI$^{13}$C, $^{14}$C, DOC, DO$^{13}$C, PO$_4^{3-}$, O$_2$ | Mook (1986) |
| PISCES | $2° \times 2°$ (mean with enhanced meridional resolution at the equator) | 30 | CaCO$_3$, CO$_3^{2-}$, DIC, $^{13}$C (in the 3 dissolved and 7 particulate carbon pools), DOC, nanophytoplankon, diatoms, mesozooplankton, microzooplankton, 2 detrital classes, PO$_4^{3-}$, NO$_3$, H$_4$SiO$_4$, Fe | Laws et al. (1995) |
| Bern3D+C | 36 cells $\times$ 36 cells | 32 | ALK, DIC, $^{13}$C, $^{14}$C, PO$_4^{3-}$, DOP[3], O$_2$, SiO$_2$, Fe | Freeman and Hayes (1992) |
| UVic | $1.8° \times 3.6°$ | 19 | ALK, DIC, $^{13}$C (in the 5 carbon pools), phytoplankton (nitrogen fixers and other phytoplankton), zooplankton, detritus, PO$_4^{3-}$, NO$_3$, O$_2$ | Popp et al. (1989) |





| Model | Horizontal resolution | Levels | Tracers | $\alpha_{POC \leftarrow aq}$ parameterisation |
|---|---|---|---|---|
| iLOVECLIM | $3° \times 3°$ | 20 | ALK, $CaCO_3$, DIC, $\Delta^{14}C$, $\delta^{13}C$, DOC, slow DOC, POC, phytoplankton, zooplankton, $PO_4^{3-}$, $NO_3$, $O_2$ | Freeman and Hayes (1992) |
| CESM | $3° \times 3°$ (development) $1° \times 1°$ (application) | 60 | ALK, $CaCO_3$, DIC, abiotic $^{14}C$ (in the 7 carbon pools), biotic $^{14}C$ (in the 7 carbon pools), $^{13}C$ (in the 7 carbon pools), DOC, diazatrophs, diatoms, small phytoplankton, zooplankton, $H_4SiO_4$ | Rau et al. (1989), Laws et al. (1995), Keller and Morel (1999) |
| CSIRO Mk3L-COAL | $1.6° \times 2.8°$ | 21 | ALK, DIC, $DI^{13}C$, $^{14}C$, general phytoplankton, diazotrophs, calcifiers, $PO_4^{3-}$, Fe, $NO_3$, $^{15}NO_3$, $N_2O$, $O_2$, abiotic $O_2$ | Constant |

[1] ALK = Alkalinity

[2] DOC = Dissolved Organic Carbon

[3] DOC = Dissolved Organic Phosphate

**Table 2**: Overview of the fractionation factors used in the sensitivity experiments.

| Simulation | $\alpha_k$ | $\alpha_{aq \leftarrow g}$, $\alpha_{DIC \leftarrow g}$ | $\alpha_p$ |
|---|---|---|---|
| *std* | Standard[1] | Variable[2] | Variable[3] |
| *ki–fract-only* | Standard | 1 | 1 |
| *no-asgx-fract* | 1 | 1 | Variable |
| *no-bio-fract* | Standard | Variable | 1 |

[1] 0.99919

[2] Calculated as per Eq. (6 – 7)

[3] With $\alpha_{POC \leftarrow aq}$ calculated as per Eq. (9)





**Figures**

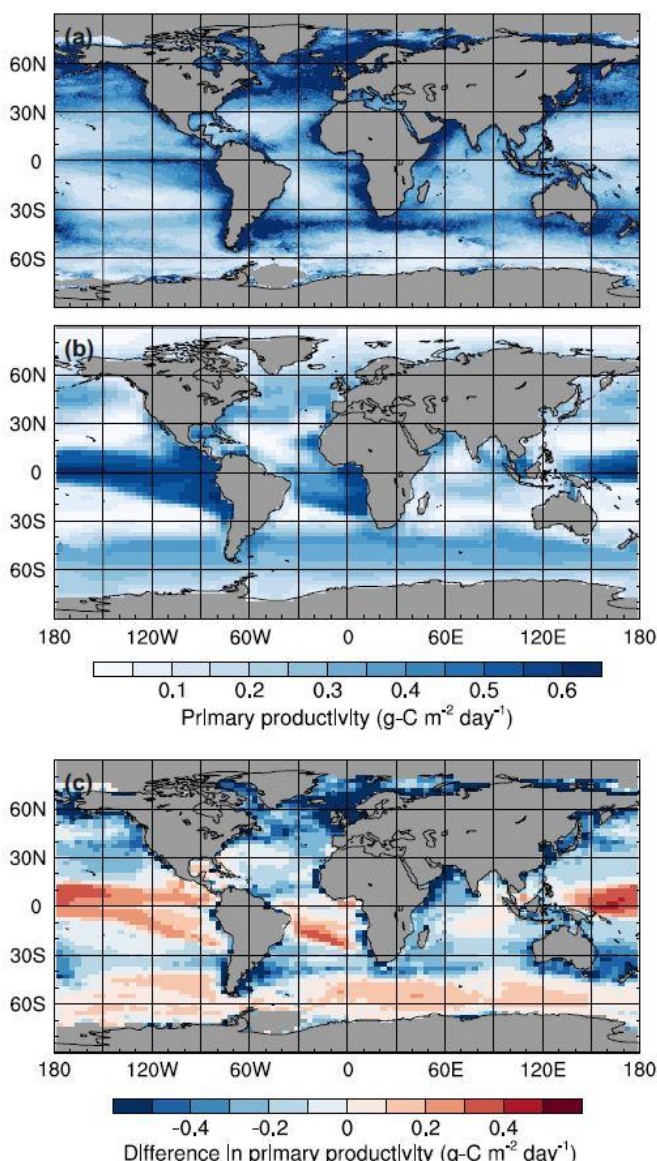

**Figure 1:** Mean annual surface primary productivity: (a) observations estimated from surface chlorophyll concentrations using the Vertically Generalised Production Model (Behrenfeld and Falkowski, 1997), (b) the *std* simulation in the 1990s, and (c) simulated minus observed. Monthly mean primary productivity data were obtained from the Oregon State University Ocean Productivity website (http://www.science.oregonstate.edu/ocean.productivity).





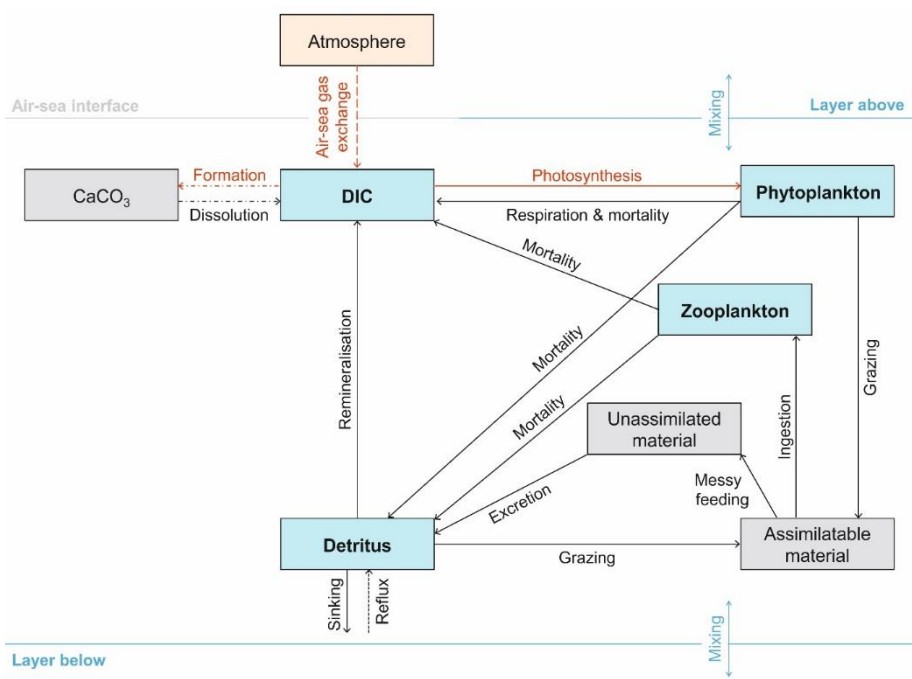

**Figure 2:** Schematic overview of the [13]C implementation in HadOCC. Blue boxes represent permanent carbon pools. Grey boxes represent temporary carbon pools. The orange box represents the prescribed atmospheric carbon pool. The dashed line represents fluxes of [13]C/[12]C. Solid lines represent fluxes of [13]C. Dot-dashed lines represent processes that occur below the lysocline ($\approx$ 2500 m below sea level). The dotted line represents the reflux of detrital material from the seafloor to the surface layer. Red lines represent fractionation effects. Note that all simulations presented in this study were run without fractionation during calcium carbonate formation ($\alpha_{CaCO3} = 1.0$).



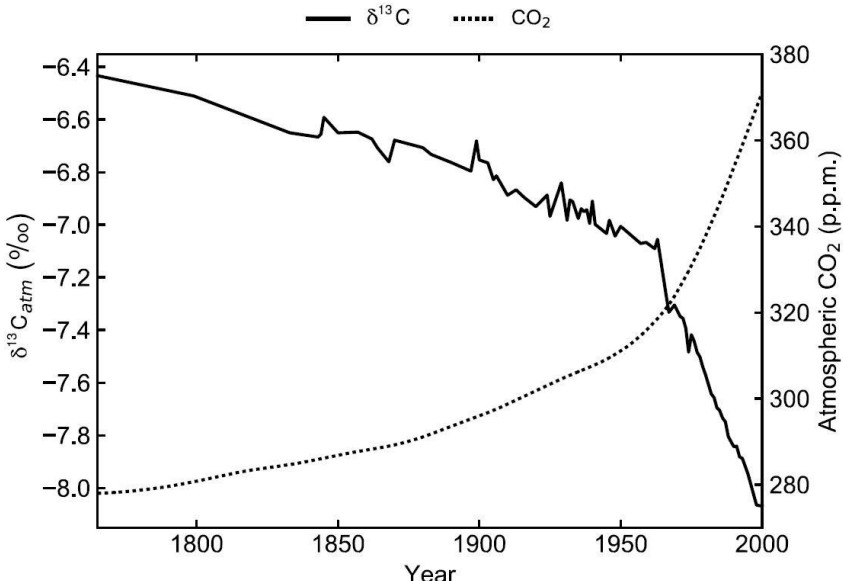

**Figure 3:** Prescribed atmospheric $\delta^{13}C$ values (solid) from the Law Dome and South Pole ice core records (Rubino et al., 2013) and prescribed atmospheric $CO_2$ values (dashed) from the OCMIP-2 files (Orr et al., 2000).





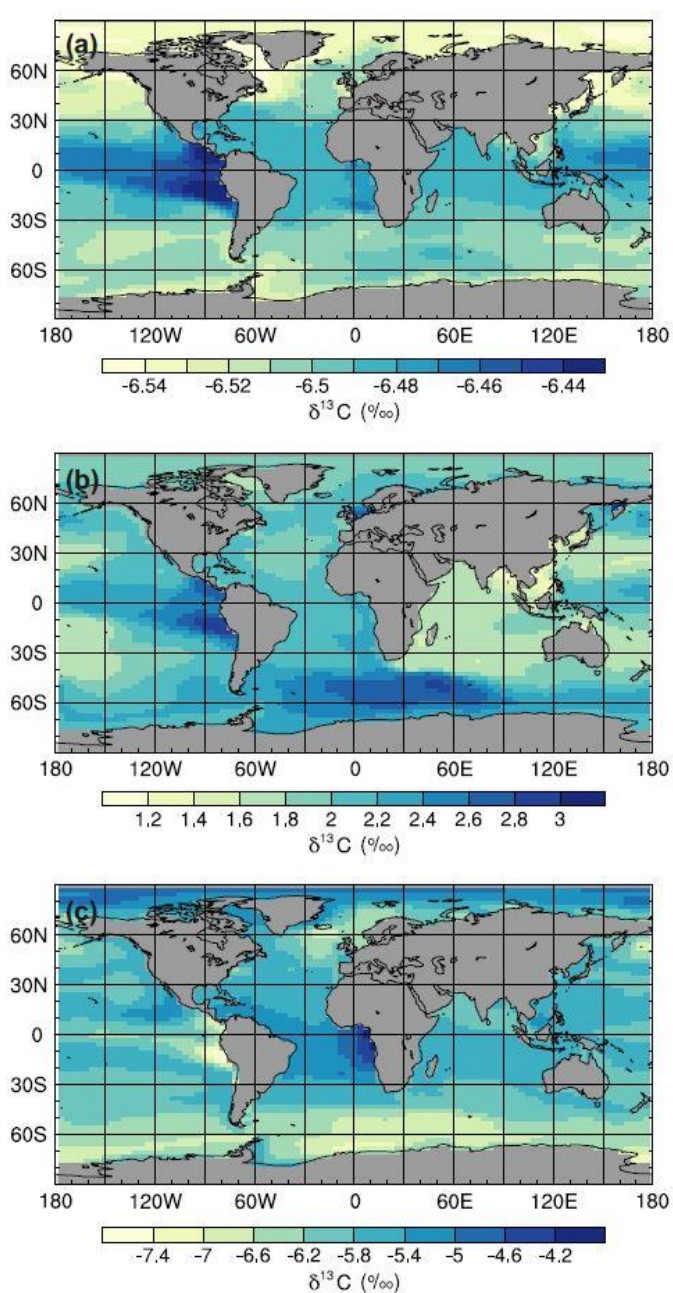

**Figure 4:** Mean annual surface $\delta^{13}C_{DIC}$ values at the end of the sensitivity experiment spin-up simulations (years 9900 to 10,000): (a) *ki-fract-only*, (b) *no-bio-fract*, and (c) *no asgx-fract*.



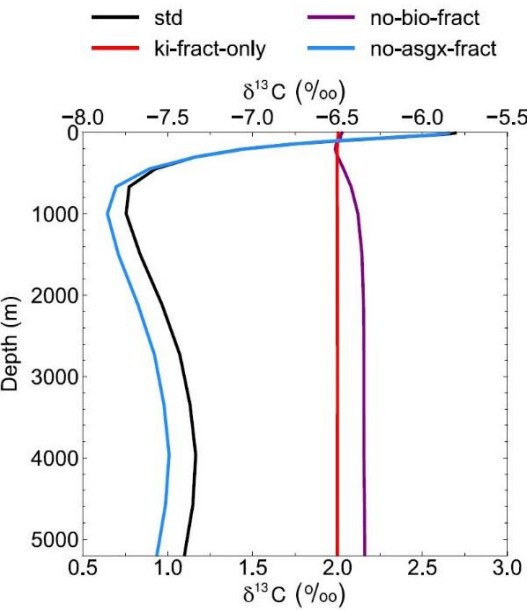

**Figure 5:** Depth profiles of globally averaged $\delta^{13}C_{DIC}$ at the end of the sensitivity experiment spin-up simulations (years 9900 to 10,000). The *std* (black) and *no-bio-fract* (purple) simulations use the bottom axis, whilst the *ki-fract-only* (red) and *no-asgx-fract* (blue) simulations use the top axis.





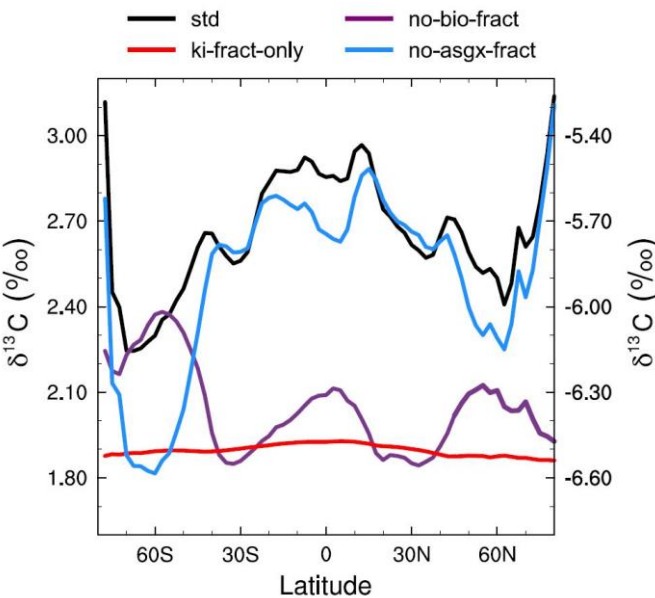

**Figure 6:** Zonally averaged mean annual surface $\delta^{13}C_{DIC}$ at the end of the sensitivity experiment spin-up simulations (years 9900 to 10,000). The *std* (black) and *no-bio-fract* (purple) simulations use the left-hand axis, whilst the *ki-fract-only* (red) and *no-asgx-fract* (blue) simulations use the right-hand axis.

**Figure 7:** Mean annual surface δ¹³C$_{DIC}$ during the 1990s: (a) observations from GLODAPv2 (Key et al., 2015; Olsen et al., 2016), (b) the *std* simulation corrected for the mean surface bias (0.97 ‰), which is calculated as ∑(simulated-observed)/number of observations, (c) the *std* simulation, (d) *std* minus GLODAPv2, (e) the *L95* simulation corrected for the mean surface bias (1.13 ‰), (f) the *L95* simulation, (g) *L95* minus GLODAPv2, (h) the *L97* simulation corrected for the mean surface bias (1.15 ‰), (i) the *L97* simulation, and (j) *L97* minus GLODAPv2.

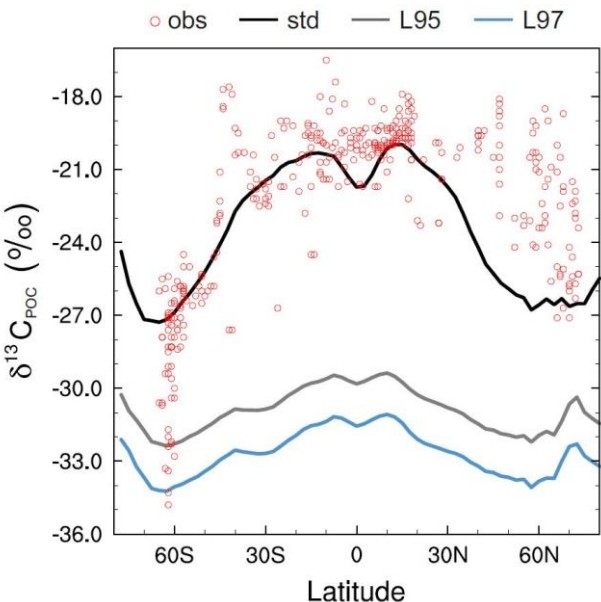

**Figure 8:** Zonally averaged mean annual mixed layer $\delta^{13}C_{POC}$ during the 1990s: observations (Goericke and Fry, 1994; red), the *std* simulation (black), the *L95* simulation (grey), and the *L97* simulation (blue).

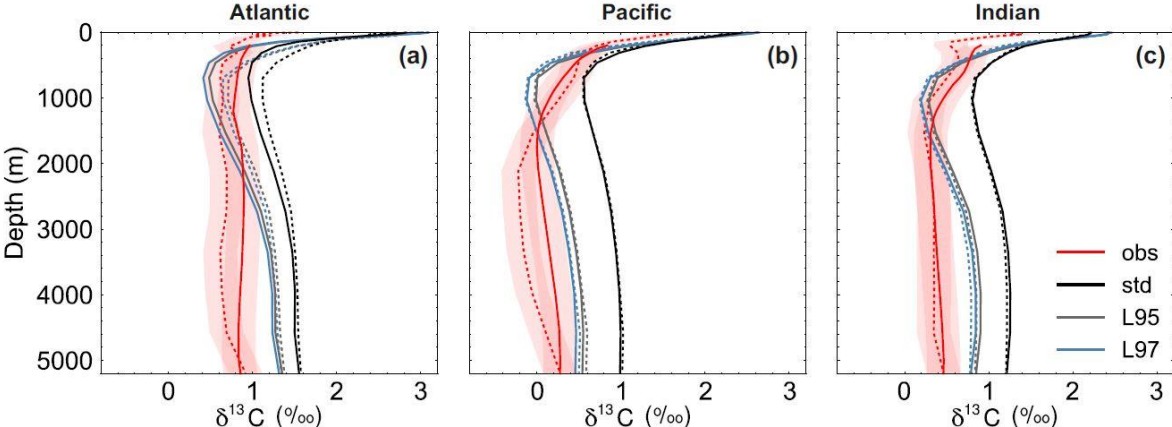

**Figure 9:** Depth profiles of $\delta^{13}C_{DIC}$ during the 1990s: (a) Atlantic Ocean, (b) Pacific Ocean, and (c) Indian Ocean. The $\delta^{13}C_{DIC}$ values in the *std* (black), *L95* (grey) and *L97* (blue) simulations are compared to observations (red). Solid lines are used for the global dataset, with observations from the gridded climatology produced by Eide et al. (2017). The simulated values have also been sub-sampled at the locations where there is a corresponding observation in the GLODAPv2 dataset (Key et al., 2015; Olsen et al., 2016; dashed). The red shading shows the estimated uncertainty in $\delta^{13}C_{DIC}$ observations due to unresolved inter-calibration between different laboratories (±0.2 ‰; Schmittner et al., 2013; Eide et al., 2017).



**Figure 10:** Zonal mean $\delta^{13}C_{DIC}$ during the 1990s in the Atlantic Ocean (left), Pacific Ocean (centre) and Indian Ocean (right): (a – c) gridded observations (Eide et al., 2017), (d – f) the *std* simulation, (g – i) the *L95* simulation, and (j – l) the *L97* simulation.



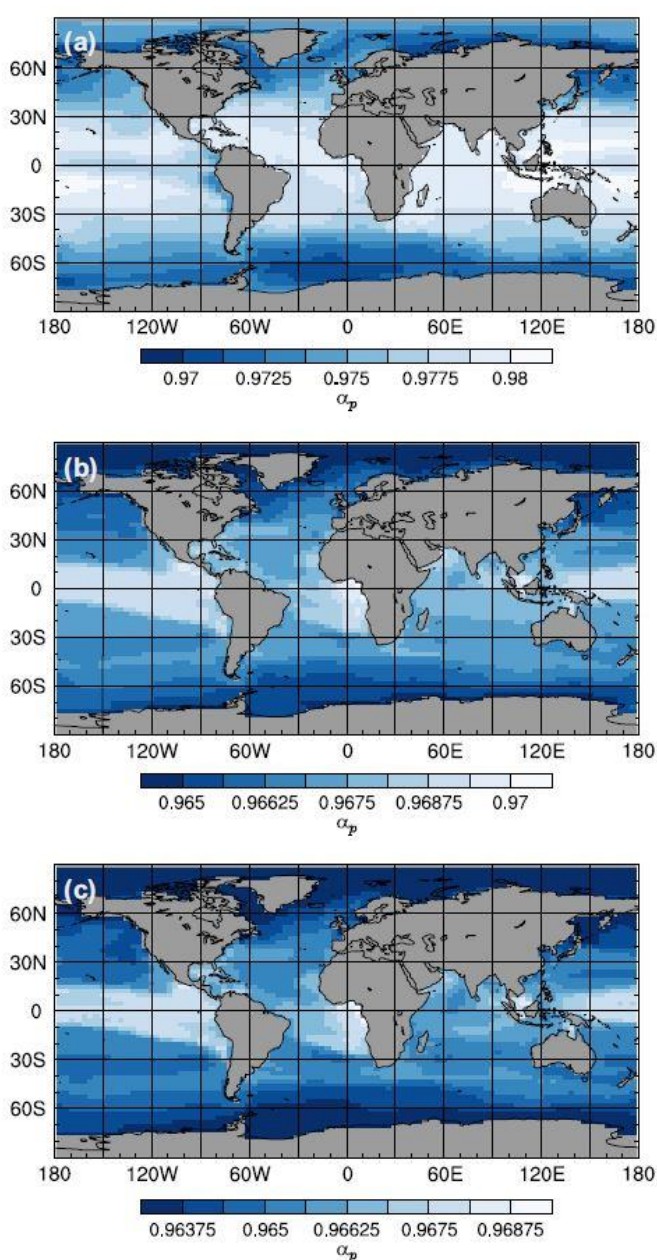

**Figure 11:** Mean annual isotopic fractionation during photosynthesis ($\alpha_p$) in the surface ocean at the end of the spin-up simulations (years 9900 to 10,000): (a) the *std* simulation, (b) the *L95* simulation, and (c) the *L97* simulation.





**Appendices**

**Appendix A: Virtual fluxes**

The standard equation for calculating the virtual flux to account for the dilution or concentration effect of surface freshwater fluxes is:

$$\frac{d^{12}C}{dt} = {}^{12}C \cdot \frac{(E-P)}{dz} \tag{A1}$$

where $E$ is evaporation, $P$ is precipitation, and $dz$ is layer depth.

As we carry $^{13}C$ as a ratio ($^{13}C/^{12}C$), virtual fluxes are not required:

$$\frac{d\left(\frac{^{13}C}{^{12}C}\right)}{dt} = \frac{{}^{12}C \cdot \frac{d^{13}C}{dt} - {}^{13}C \cdot \frac{d^{12}C}{dt}}{\left({}^{12}C\right)^2} \tag{A2}$$

$$\frac{d\left(\frac{^{13}C}{^{12}C}\right)}{dt} = \frac{1}{^{12}C} \cdot \left[{}^{13}C \cdot \frac{(E-P)}{dz}\right] - \frac{^{13}C}{\left({}^{12}C\right)^2} \cdot \left[{}^{12}C \cdot \frac{(E-P)}{dz}\right] \tag{A3}$$

$$\frac{d\left(\frac{^{13}C}{^{12}C}\right)}{dt} = 0 \tag{A4}$$

**Appendix B: Air-sea gas exchange equations**

The standard equation for calculating the change in DI$^{13}$C due to air-sea gas exchange is:

$$\frac{d^{13}C}{dt} = \alpha_k \cdot \alpha_{aq \leftarrow g} \cdot PV \cdot \left( C_{sat} \cdot \frac{^{13}A}{^{12}A} - \frac{C_{surf} \cdot \frac{^{13}C}{^{12}C}}{\alpha_{DIC \leftarrow g}} \right) \tag{B1}$$

where $PV$ is the piston velocity (Eq. (3)), $C_{sat}$ is the saturation concentration of atmospheric $CO_2$ (in mol m$^{-3}$), $C_{surf}$ is the surface aqueous concentration of $CO_2$ (in mol m$^{-3}$), $\alpha_k$ is the constant kinetic fractionation factor, $\alpha_{aq \leftarrow g}$ is the temperature-dependent fractionation during gas dissolution (Eq. (6)), $\alpha_{DIC \leftarrow g}$ is the is the temperature-dependent fractionation between aqueous $CO_2$ and DIC (Eq. (7)), and $^{13}A/^{12}A$ and $^{13}C/^{12}C$ are the $^{13}C/^{12}C$ ratios of the atmosphere and DIC, respectively.

The equation for calculating the change in DI$^{13}$C/ DI$^{12}$C due to air-sea gas exchange is:

$$\frac{d\left(\frac{^{13}C}{^{12}C}\right)}{dt} = \frac{{}^{12}C \cdot \frac{d^{13}C}{dt} - {}^{13}C \cdot \frac{d^{12}C}{dt}}{\left({}^{12}C\right)^2} \tag{B2}$$

$$\frac{d\left(\frac{^{13}C}{^{12}C}\right)}{dt} = \frac{1}{^{12}C} \cdot \left[ \alpha_k \cdot \alpha_{aq \leftarrow g} \cdot PV \cdot \left( C_{sat} \cdot \frac{^{13}A}{^{12}A} - \frac{C_{surf} \cdot \frac{^{13}C}{^{12}C}}{\alpha_{DIC \leftarrow g}} \right) \right] - \frac{^{13}C}{\left({}^{12}C\right)^2} \cdot \left[ PV \cdot (C_{sat} - C_{surf}) \right] \tag{B3}$$





$$\frac{d\left(\frac{^{13}C}{^{12}C}\right)}{dt} = \frac{1}{^{12}C} \cdot PV \cdot \left[ \alpha_k \cdot \alpha_{aq\leftarrow g} \cdot \left( C_{sat} \cdot \frac{^{13}A}{^{12}A} - \frac{C_{surf} \cdot \frac{^{13}C}{^{12}C}}{\alpha_{DIC\leftarrow g}} \right) - \left( \frac{^{13}C}{^{12}C} \cdot [C_{sat} - C_{surf}] \right) \right] \qquad (B4)$$

**Appendix C: Biological equations**

For consistency with the standard biological tracers, the $^{13}$C contents of phytoplankton ($^{13}$P), zooplankton ($^{13}$Z) and detritus ($^{13}$D) are carried in mmol-N m$^{-3}$, with the carbon concentrations and fluxes calculated using fixed stoichiometric ratios. The DI$^{13}$C/DI$^{12}$C values are therefore converted from a ratio in model units (Eq. (1)) to normalised DI$^{13}$C concentrations before entering the soft tissue pump. The conversion is reversed at the end of each timestep.

**C.1 Phytoplankton (P)**

The standard equation for calculating the change in phytoplankton ($^{12}$P) is:

$$\frac{dP}{dt} = R_P - G_p - m_P - \eta_P \qquad (C1)$$

where $R_P$ is the specific growth rate of phytoplankton, $G_p$ represents grazing by zooplankton, $m_P$ represents phytoplankton mortality due to overpopulation, and $\eta_P$ represents phytoplankton respiration.

The equation for calculating the change in $^{13}$P is:

$$\frac{d^{13}P}{dt} = R_P \times \frac{^{13}C}{^{12}C} \times \alpha_p - G_p \times \frac{^{13}P}{^{12}P} - m_P \times \frac{^{13}P}{^{12}P} - n_P \times \frac{^{13}P}{^{12}P} \qquad (C2)$$

where $\alpha_p$ is the isotopic fractionation that occurs during photosynthesis (Eq. (8)), $^{13}C/^{12}C$ is the $^{13}$C/$^{12}$C ratio of DIC, and $^{13}P/^{12}P$ is the $^{13}$C/$^{12}$C ratio of phytoplankton.

The $^{13}$P tracer is updated using the forward Euler method:

$$^{13}P_{(t+\Delta t)} = {}^{13}P_{(t)} + \Delta t \times \left( R_{P(t)} \times \frac{^{13}C}{^{12}C}_{(t)} \times \alpha_{p(t)} - G_{p(t)} \times \frac{^{13}P}{^{12}P}_{(t)} - m_{P(t)} \times \frac{^{13}P}{^{12}P}_{(t)} - n_{P(t)} \times \frac{^{13}P}{^{12}P}_{(t)} \right) \qquad (C3)$$

**C.2 Zooplankton (Z)**

The standard equation for calculating the change in zooplankton ($^{12}$Z) is:

$$\frac{dZ}{dt} = \beta_P \times G_P + \beta_D \times G_D - m_Z \qquad (C4)$$

where $\beta_P$ and $\beta_D$ are the assimilation efficiencies associated with zooplankton grazing on phytoplankton ($G_P$) and detritus ($G_D$), respectively, and $m_Z$ represents zooplankton mortality due to predation and natural causes.

The equation for calculating the change in $^{13}$Z is:

$$\frac{d^{13}Z}{dt} = \beta_P \times G_P \times \frac{^{13}P}{^{12}P} + \beta_D \times G_D \times \frac{^{13}D}{^{12}D} - m_Z \times \frac{^{13}Z}{^{12}Z} \qquad (C5)$$

where $^{13}P/^{12}P$, $^{13}D/^{12}D$ and $^{13}Z/^{12}Z$ are the isotopic ratios of phytoplankton, detritus and zooplankton, respectively.



The $^{13}Z$ tracer is updated using the forward Euler method:

$$^{13}Z_{(t+\Delta t)} = {}^{13}Z_{(t)} + \Delta t \times \left( \beta_{P(t)} \times G_{P(t)} \times \frac{^{13}P}{^{12}P}_{(t)} + \beta_{D(t)} \times G_{D(t)} \times \frac{^{13}D}{^{12}D}_{(t)} - m_{Z(t)} \times \frac{^{13}Z}{^{12}Z}_{(t)} \right) \tag{C6}$$

**C.3 Dissolved inorganic carbon (DIC, C)**

The standard equation for calculating the change in DI$^{12}$C is:

$$\frac{dC}{dt} = -R_P + \lambda_D + (1 - \beta_P) \times G_p + (1 - \beta_D) \times G_D + m_Z + m_P + \eta_P \tag{C7}$$

where $R_P$ is the specific growth rate of phytoplankton, $\lambda_D$ is detrital remineralisation, which is specified at a constant rate (0.1 day$^{-1}$ in the uppermost 250 m of the ocean and 0.02 day$^{-1}$ at all other depths), $\beta_P$ and $\beta_D$ are the assimilation efficiencies associated with zooplankton grazing on phytoplankton ($G_P$) and detritus ($G_D$), respectively, $m_Z$ represents zooplankton mortality due to predation and natural causes, $m_P$ represents phytoplankton mortality due to overpopulation, and $\eta_P$ represents phytoplankton respiration.

The equation for calculating the change in DI$^{13}$C is:

$$\frac{d^{13}C}{dt} = -R_P \times \frac{^{13}C}{^{12}C} \times \alpha_p + \lambda_D \times \frac{^{13}D}{^{12}D} + (1 - \beta_P) \times G_p \times \frac{^{13}P}{^{12}P} + (1 - \beta_D) \times G_D \times \frac{^{13}D}{^{12}D} + m_Z \times \frac{^{13}Z}{^{12}Z} + m_P \times \frac{^{13}P}{^{12}P} + \eta_P \times \frac{^{13}P}{^{12}P}$$

$$\tag{C8}$$

where $\alpha_p$ is the isotopic fractionation that occurs during photosynthesis (Eq. (8)) and $^{13}C/^{12}C$, $^{13}D/^{12}D$, $^{13}P/^{12}P$ and $^{13}Z/^{12}Z$ are the isotopic ratios of DIC, detritus, phytoplankton and zooplankton, respectively.

The DI$^{13}$C tracer is updated using the forward Euler method:

$$^{13}C_{(t+\Delta t)} = {}^{13}C_{(t)} + \Delta t \times \left( -R_{P(t)} \times \frac{^{13}C}{^{12}C}_{(t)} \times \alpha_{p(t)} + \lambda_{D(t)} \times \frac{^{13}D}{^{12}D}_{(t)} + \left(1 - \beta_{P(t)}\right) \times G_{p(t)} \times \frac{^{13}P}{^{12}P}_{(t)} + \left(1 - \beta_{D(t)}\right) \times \right.$$

$$\left. G_{D(t)} \times \frac{^{13}D}{^{12}D}_{(t)} + m_{Z(t)} \times \frac{^{13}Z}{^{12}Z}_{(t)} + m_{P(t)} \times \frac{^{13}P}{^{12}P}_{(t)} + \eta_{P(t)} \times \frac{^{13}P}{^{12}P}_{(t)} \right) \tag{C9}$$

**C.4 Detritus (D)**

Unlike the other biological tracers, the standard detritus tracer ($^{12}$D) is updated using a semi-implicit scheme:

$$\frac{D_{(t+\Delta t,k)} - D_{(t,k)}}{\Delta t} = \frac{dD}{dt}_{bio(t,k)} + \frac{dD}{dt}_{sink\_in(t+\Delta t,k-1)} - \frac{dD}{dt}_{sink\_out(t+\Delta t,k)} \tag{C10}$$

$$D_{(t+\Delta t,k)} - D_{(t,k)} = \Delta t \times D_{bio(t,k)} + \Delta t \times \frac{\gamma}{dz/100} \times D_{(t+\Delta t,k-1)} - \Delta t \times \frac{\gamma}{dz/100} \times D_{(t+\Delta t,k)} \tag{C11}$$

$$D_{(t+\Delta t,k)} + \Delta t \times \frac{\gamma}{dz/100} \times D_{(t+\Delta t,k)} = D_{(t,k)} + \Delta t \times D_{bio(t,k)} + \Delta t \times \frac{\gamma}{dz/100} \times D_{(t+\Delta t,k-1)} \tag{C12}$$

$$D_{(t+\Delta t,k)} \times \left(1 + \Delta t \times \frac{\gamma}{dz/100}\right) = D_{(t,k)} + \Delta t \times D_{bio(t,k)} + \Delta t \times \frac{\gamma}{dz/100} \times D_{(t+\Delta t,k-1)} \tag{C13}$$





$$D_{(t+\Delta t,k)} = \frac{D_{(t,k)} + \Delta t \times D_{bio(t,k)} + \Delta t \times \frac{\gamma}{dz/100} \times D_{(t+\Delta t,k-1)}}{1 + \Delta t \times \frac{\gamma}{dz/100}} \tag{C14}$$

$$D_{(t+\Delta t,k)} = D_{(t,k)} + \frac{dD}{dt}_{(t,k)} \tag{C15}$$

$$\frac{dD}{dt}_{(t,k)} = D_{(t+\Delta t,k)} - D_{(t,k)} \tag{C16}$$

$$\frac{dD}{dt}_{(t,k)} = \frac{D_{(t,k)} + \Delta t \times D_{bio(t,k)} + \Delta t \times \frac{\gamma}{dz/100} \times D_{(t+\Delta t,k-1)}}{1 + \Delta t \times \frac{\gamma}{dz/100}} - D_{(t,k)} \tag{C17}$$

where $t$ is the current timestep, $k$ is the model level, $dD/dt_{bio}$ is the change in detritus due to biological effects (Eq. (C19)), $\gamma$ is the sinking rate, which is parameterised at 10 m day$^{-1}$, $dz$ is the depth of the layer (in cm), and $D$ is the detritus concentration. Following the same principles, the $^{13}$D tracer is updated using:

$$\frac{d^{13}D}{dt}_{(t,k)} = \frac{^{13}D_{(t,k)} + \Delta t \times {^{13}D}_{bio(t,k)} + \Delta t \times \frac{\gamma}{dz/100} \times {^{13}D}_{(t+\Delta t,k-1)}}{1 + \Delta t \times \frac{\gamma}{dz/100}} - {^{13}D}_{(t,k)} \tag{C18}$$

### C.4.1 Biological effects

The standard equation for calculating the change in detritus ($^{12}$D) due to biology is:

$$\frac{dD}{dt}_{bio} = m_Z + m_P - \lambda_D - G_D - (1-\beta_P) \times G_p - (1-\beta_D) \times G_D \tag{C19}$$

where $m_Z$ represents zooplankton mortality due to predation and natural causes, $m_P$ represents phytoplankton mortality due to overpopulation, $\lambda_D$ is detrital remineralisation, which is specified at a constant rate (0.1 day$^{-1}$ in the uppermost 250 m of the ocean and 0.02 day$^{-1}$ at all other depths), and $\beta_P$ and $\beta_D$ are the assimilation efficiencies associated with zooplankton grazing

on phytoplankton ($G_P$) and detritus ($G_D$), respectively.

The equation for calculating the change in $^{13}$D due to biology is:

$$\frac{d^{13}D}{dt}_{bio} = m_Z \times \frac{^{13}Z}{^{12}Z} + m_P \times \frac{^{13}P}{^{12}P} - \lambda_D \times \frac{^{13}D}{^{12}D} - G_D \times \frac{^{13}D}{^{12}D} - (1-\beta_P) \times G_p \times \frac{^{13}P}{^{12}P} - (1-\beta_D) \times G_D \times \frac{^{13}D}{^{12}D} \tag{C20}$$

where $^{13}Z/^{12}Z$, $^{13}P/^{12}P$ and $^{13}D/^{12}D$ are the isotopic ratios of zooplankton, phytoplankton, and detritus, respectively.

### C.4.2 Reflux

The small amount of detritus that reaches the ocean floor is immediately refluxed back to the surface layer to conserve nitrogen and carbon.

$$\frac{dD}{dt}_{sink\_in(k=1)} = \frac{\gamma}{dz/100} \cdot D_{(k=KMT)} \tag{C21}$$

where $k$ is the model level, $\gamma$ is the sinking rate, which is parameterised at 10 m day$^{-1}$, $dz$ is the depth of the layer (in cm), $D$ is the detritus concentration, and $KMT$ is the maximum depth of the ocean. The same equation applies for $^{13}$D.