# Peer review of "Simulating stable carbon isotopes in the ocean component of the FAMOUS General Circulation Model with MOSES1 (XOAVI)"

_Geoscientific Model Development, 2019_

## Short Comment (SC1) · 23 Jan 2020

Dear authors,

in my role as Executive editor of GMD, I would like to bring to your attention our Editorial version 1.2:

https://www.geosci-model-dev.net/12/2215/2019/

This highlights some requirements of papers published in GMD, which is also available on the GMD website in the 'Manuscript Types' section:

http://www.geoscientific-model-development.net/submission/manuscript_types.html

[Figure]

In particular, please note that for your paper, the following requirement has not been met in the Discussions paper:

- "The main paper must give the model name and version number (or other unique identifier) in the title."

Please add a version numbers for FAMOUS to the title upon your revised submission to GMD.

Yours,

Astrid Kerkweg

---

## Referee Comment (RC1) · Anonymous Referee #1 · 8 Feb 2020

Dentith et al., present the implementation of oceanic stable carbon isotope (13C) in the ocean component of the FAMOUS model. The model includes carbon fractionation during air-sea gas exchange, and photosynthesis. Three schemes are tested for fractionation during photosynthesis, with the more complex schemes not improving the model-data comparison. The oceanic d13C values are globally higher in the model than in observations, probably due to the representation of both the oceanic circulation and marine carbon cycle in the model. I agree with the conclusions of the authors that the model should probably be re-tuned for both its oceanic circulation and marine carbon cycle. But, before that another quick check of d13C implementation could be done. I support publication of the manuscript once the comments below have been

[Figure]

taken into account.

P3, L. 5-8 and table 1: the carbon isotopes enabled model LOVECLIM is missing from this list, with references Mouchet, 2013 (Radiocarbon) and Menviel et al., 2015 (GBC), as the carbon cycle models and carbon isotopes implementation are different in iLOVECLIM and LOVECLIM.

P3, L9-11: This sentence has to be amended. The list of carbon isotopes enabled models on L. 8 includes 2-dimensional, 3-dimensional models, OGCMs and AOGCMs; and apart from 3 there are OGCMs with similar resolution as the model described here. It thus cannot be suggested that all these models are too simple to study abrupt climate changes, moreover when the model presented in this study probably provides similar performances/capability to some models in that list (even more so because it focuses on ocean processes, and because in this model "sea ice formation and melt do not affect salinity distributions", p5, L. 16).

P5, L. 14: suggest to replace "EEP", by "equatorial Pacific" as simulated primary pro­duction is higher both in the western and eastern Pacific. Please also modify the end of the sentence as follows: "attributed to excessive upwelling in the EEP".

From P10, L. 28 to p11, L.2: I would suggest to be really cautious here and eventu­ally add a few words/sentences of explanation as it is stated that d13C is high in the Southern Ocean because of CO2 outgassing, and high in the EEP because AABW is upwelled. This is of course only true in this experiment where no biological fractionation is taken into account. If all processes are taken into account AABW would have a low d13C. Therefore, to account for the quick reader, it might be best to repeat "because there is no biological fractionation in this experiment". In addition, I doubt that "AABW" is upwelled in the EEP. It is sufficient to simply state that the upwelled water has a high d13C value because it is mostly southern-sourced waters. But, it is quite surprising to see such high values in the EEP in no-bio-fract. In addition, the model d13C is globally too high in std. The authors investigate thoroughly the impact of fractionation during

photosynthesis, but could it be a problem linked to gas-exchange parametrization? Is the right hand side of Equation 5 really needed? Would plotting d13C vs PO4 in the model help in confirming that everything is correct?

Figure 6 is quite helpful. The profiles are a bit surprising, but are discussed in details and compared with Schmittner et al. 2013. Note that the Bern3D and LOVECLIM profiles are also shown in Menviel et al., 2015, and could help in assessing the accuracy of the latitudinal d13C distributions.

Figure 3: Why is CaCO3 in grey? Since there is no fractionation during CaCO3 formation, then maybe the line from DIC to CaCO3 should be black. There should be an arrow going from DIC to atm. "Atmosphere" should in fact be "Atm CO2".

Figure 7: The authors might consider modifying this figure as it looks like 3 times the same plot. I understand that it might be the point, but maybe best to move some (L95,L97) to SI. The choice of the colorbars could be revised: it is hard (impossible) to see any feature in a), and difficult in the right hand side panels. Are experiments L95 and L97 described in the methods? It would be helpful to also include them in Table 2.

---

## Referee Comment (RC2) · Anonymous Referee #2 · 13 Feb 2020

General comments

This paper presents the inclusion of stable carbon isotopes, i.e. 13C, in the FAMOUS model. The authors have evaluated the effect of fractionation by air-sea gas exchange, biology and ocean circulation, and have tested different fractionation parameterisations for biology. This is a very useful development of the model that will be very valuable for paleo studies. This work is well presented and the reasoning is easy to follow. It has already demonstrated its usefulness by showing that the discrepancy between model results and data is likely to be due to biases in the simulated climate as well as the biogeochemical model.

[Figure]

My main concern is that this model should not be used as it is for paleo studies but should be re-tuned, especially the biogeochemical module, as there are very large disagreements between simulated d13C and data. This is highlighted by the authors and re-tuning the model is clearly out of the scope of this study, but it might nonetheless be interesting to have a few additional sensitivity simulations to evaluate how much the results could be improved if the biogeochemical model was slightly modified, for example with a modified remineralisation, which could potentially help reduce the model-data disagreements.

Specific comments

Abstract

p.1 l. 7: do you mean "carbon isotopic ratios" instead of isotopic ratios?

Introduction

p.2 l.2. The first sentence is almost the same as the first sentence from the abstract: maybe change it?

p.2. l.2-10. This is not entirely true for 14C as it also depends on radioactive decay: maybe you could say right at the beginning (after giving the percentages) that C14 is not studied here and only keep 12C and 13C in this part.

p2. l.9. You could give the complete definition of d13C mode explicitely as this is entirely on d13C inclusion it is worth reminding clearly the definition (d13C=...).

p.3 l. 5-8. You should also include the LOVECLIM model. What about Genie?

p.3 l.19. There is an arrow that should be deleted between "studying" and "complex" .

p5. Line 16. Sea ice does not change salinity: this is out of the scope of the study but probably needs to be modified in the model...

P5. Line 27-28. This seems at odds with what is said later. From what I understand

from this paper both the physical model AND the biogeochemical model are responsible for carbon isotopes mismatch between simulation results and data and disentangling between the two is not done here.

Results and discussion

p. 10 l. 15 / Figure 4. I would start with standard results before looking at the sensitivity experiments to be able to compare these sensitivity experiments with the standard one. So, on Figure 4 I would add the standard simulated d13C first as (a) and then the other 3 sensitivity simulations as b-d, which would also be more coherent with having the 4 simulations on Figure 5.

p.10 line 19. Is this a simulation that you actually did to verify this or just discussion? Please specify.

p.11 line 8. Could you quickly remind the reader what this simulation is (to avoid looking for it earlier in the text)?

p.14 l.18. Could you test your hypothesis for the cause of the model-data mismatch due to the export ratio and remineralisation rate vs biases in ocean circulation by running additional sensitivity experiments? Testing the ocean circulation is probably more difficult, but modifying the export ratio and/or remineralisation to evaluate if this could have a large contribution to the mismatch is probably easier.

---

## Author Response (AR1)

*We would like to thank the two anonymous reviewers and the executive editor for their feedback on our manuscript. We have considered all of the comments carefully and addressed them in turn below, with our responses in blue-italics. In all of our responses, page numbers and line numbers refer to the revised manuscript.*

*We note that both reviewers suggested only minor revisions to the manuscript, therefore we have made the following changes:*

- *Extra examples of $^{13}$C-enabled models added to Table 1*
- *Minor edits to figures*
- *Wordsmithing for clarity and adding in additional details (where requested)*
* * *
**Executive editor**

Dear authors,

in my role as Executive editor of GMD, I would like to bring to your attention our Editorial version 1.2:
https://www.geosci-model-dev.net/12/2215/2019/

This highlights some requirements of papers published in GMD, which is also available on the GMD website in the 'Manuscript Types' section: http://www.geoscientific-model-development.net/submission/manuscript_types.html

In particular, please note that for your paper, the following requirement has not been met in the Discussions paper:

"The main paper must give the model name and version number (or other unique identifier) in the title."

Please add a version numbers for FAMOUS to the title upon your revised submission to GMD.

Yours,

Astrid Kerkweg

*The unique identifier for the FAMOUS code and setup used in this manuscript is already included in the title of our manuscript ('XOAVI'), enabling readers to ascertain, reference and access/use the exact combination of code, model components, and inputs as is presented here. We realise that this does not follow a more conventional versioning (e.g. linear) format, such as 'FAMOUS v1.1', however, no such versioning exists for FAMOUS (see other FAMOUS publications in the GMD special issue). To give further details (and as outlined in Table 3): XOAVI is the unique identifier for the standard transient simulation, which forms the basis of our discussions and the core of all our simulations. The unique identifiers for all of the other sensitivity simulations presented in our manuscript are also outlined in Table 3, should a reader wish to repeat any or all of these simulations in the future. In any case, the unique identifier is already in the title, as required, therefore no changes have been made in the revised manuscript.*

**Reviewer #1**

Dentith et al., present the implementation of oceanic stable carbon isotope ($^{13}$C) in the ocean component of the FAMOUS model. The model includes carbon fractionation during air-sea gas exchange, and photosynthesis. Three schemes are tested for fractionation during photosynthesis, with the more complex schemes not improving the model-data comparison. The oceanic $\delta^{13}$C values are globally higher in the model than in observations, probably due to the representation of both the oceanic circulation and marine carbon cycle in the model. I agree with the conclusions of the authors that the model should probably be re-tuned for both its oceanic circulation and marine carbon cycle. But, before that another quick check of $\delta^{13}$C implementation could be done. I support publication of the manuscript once the comments below have been taken into account.

*We are confident that our implementation is mathematically correct and that there are no bugs in the code because we have already completed extensive checks, including:*

- *Verifying that our equations are balanced, and that no carbon ($^{12}$C or $^{13}$C) is being created or destroyed.*
- *Running with $\delta^{13}C_{atm}$ equal to 0 ‰ and no isotopic fractionation effects (all α values set to 1.0). In this simulation, the $\delta^{13}C_{DIC}$ values remained constant at 0 ‰.*
- *Running with $\delta^{13}C_{atm}$ equal to -6.5 ‰ and no isotopic effects (all α values set to 1.0). In this simulation, the $\delta^{13}C_{DIC}$ equilibrated at -6.5 ‰.*

P3, L. 5-8 and table 1: the carbon isotopes enabled model LOVECLIM is missing from this list, with references Mouchet, 2013 (Radiocarbon) and Menviel et al., 2015 (GBC), as the carbon cycle models and carbon isotopes implementation are different in iLOVECLIM and LOVECLIM.

*Our intention was to provide illustrative examples of $^{13}$C-enabled models across a range of complexities as opposed to a complete list of all $^{13}$C-enabled models, but we have added this additional example to Table 1 (p. 26 – 27).*

P3, L9-11: This sentence has to be amended. The list of carbon isotopes enabled models on L. 8 includes 2-dimensional, 3-dimensional models, OGCMs and AOGCMs; and apart from 3 there are OGCMs with similar resolution as the model described here. It thus cannot be suggested that all these models are too simple to study abrupt climate changes, moreover when the model presented in this study probably provides similar performances/capability to some models in that list (even more so because it focuses on ocean processes, and because in this model "sea ice formation and melt do not affect salinity distributions", p5, L. 16).

*FAMOUS is a full-complexity, ocean-atmosphere General Circulation Model (AOGCM). Even though it does not have the high resolution of the more complex models that are cited (e.g. PISCES and CESM), for simulations of coupled ocean-atmosphere interactions, and particularly when atmospheric variability is important, FAMOUS is an improved model compared to the Earth System Models of Intermediate Complexity Models (that do not have a full-primitive equation atmosphere or that have more limited vertical resolution in the atmosphere) and the ocean-only models. However, these are specific cases and we agree that the highlighted sentence made a crude point that was not well justified. We have therefore removed this sentence from the manuscript.*

P5, L. 14: suggest to replace "EEP", by "equatorial Pacific" as simulated primary production is higher both in the western and eastern Pacific. Please also modify the end of the sentence as follows: "attributed to excessive upwelling in the EEP".

*We have revised this sentence to "However, primary production is higher than observed in the equatorial Pacific, which is attributed to excessive upwelling in the eastern equatorial Pacific (Palmer and Totterdell, 2001)" (p. 5, l. 13 – 14.).*

From P10, L. 28 to p11, L.2: I would suggest to be really cautious here and eventually add a few words/sentences of explanation as it is stated that $\delta^{13}C$ is high in the Southern Ocean because of $CO_2$ outgassing, and high in the EEP because AABW is upwelled. This is of course only true in this experiment where no biological fractionation is taken into account. If all processes are taken into account AABW would have a low $\delta^{13}C$. Therefore, to account for the quick reader, it might be best to repeat "because there is no biological fractionation in this experiment". In addition, I doubt that "AABW" is upwelled in the EEP. It is sufficient to simply state that the upwelled water has a high $\delta^{13}C$ value because it is mostly southern-sourced waters. But, it is quite surprising to see such high values in the EEP in no-bio-fract. In addition, the model $\delta^{13}C$ is globally too high in std. The authors investigate thoroughly the impact of fractionation during photosynthesis, but could it be a problem linked to gas-exchange parametrization? Is the right hand side of Equation 5 really needed? Would plotting $\delta^{13}C$ vs $PO_4$ in the model help in confirming that everything is correct?

*We have revised the manuscript as suggested:*
*"When both the equilibrium and kinetic fractionation effects are included during air-sea gas exchange (no-bio-fract), the large-scale $\delta^{13}C_{DIC}$ distributions are closely related to the SST patterns because of the temperature dependence of $\alpha_{aq \leftarrow g}$ and $\alpha_{DIC \leftarrow g}$ (Figure 4b). In the absence of biological fractionation, relatively high $\delta^{13}C_{DIC}$ values (> +2.5 ‰) are simulated in the Southern Ocean due to the combined effect of $CO_2$ outgassing and low SSTs, both of which cause $^{13}C$ enrichment. The $\delta^{13}C_{DIC}$ values in the Arctic Ocean are comparably low because the model has more extensive sea ice in the Northern Hemisphere than in the Southern Hemisphere, which inhibits air-sea gas exchange. The highest values (+3.00 ‰) are simulated in the eastern equatorial Pacific where there are high rates of net $CO_2$ outgassing and southern-sourced waters, which have a high $\delta^{13}C_{DIC}$ signature in this simulation because there is no biological fractionation, are upwelled." (p. 10, l. 21 – 28.)*

*The right hand side of the equation 5 is necessary because we are carrying the tracer as a ratio ($^{13}C/^{12}C$). Please see the derivation in Appendix B.*

*Plotting the simulated $\delta^{13}C$ values against the corresponding $PO_4^{3-}$ values, which we have derived using Redfield ratios because FAMOUS only contains a single nitrogenous nutrient, suggests that everything is correct. The model captures the expected relationship between the two variables, with approximately a -1 ‰ change in $\delta^{13}C$ per 1 µmol kg$^{-1}$ change in $PO_4^{3-}$ (Figure R1).*

[Figure]

***Figure R1:*** *δ¹³C versus PO₄³⁻ during the 1990s in the std simulation (top) and in the GLODAPv.2 data set (bottom). The red lines are the linear regression lines for each set of data.*

Figure 6 is quite helpful. The profiles are a bit surprising, but are discussed in details and compared with Schmittner et al. 2013. Note that the Bern3D and LOVECLIM profiles are also shown in Menviel et al., 2015, and could help in assessing the accuracy of the latitudinal d13C distributions.

*Menviel et al. (2015) conducted a different set of sensitivity experiments to those presented by us (section 3.1) and Schmittner et al. (2013). Instead, they conducted a suite of experiments with freshwater forcing in the North Atlantic and Southern Oceans, and changes in the wind stress. Thus, whilst interesting, their results are not comparable to ours.*

*Since our original manuscript submission, we have accessed the raw data for the four equivalent simulations conducted by Schmittner et al. (2013), which we discuss in section 3.1. We have therefore added these lines on to Figure 5 and Figure 6 so that readers can make a direct comparison between the two models. The figures and the corresponding captions have therefore be revised as follows:*

[Figure]

*Figure 5: Depth profiles of globally averaged $\delta^{13}C_{DIC}$ at the end of the sensitivity experiment spin-up simulations (years 9900 to 10,000). The std (black) and no-bio-fract (purple) simulations use the bottom axis, whilst the ki-fract-only (red) and no-asgx-fract (blue) simulations use the top axis. The dotted lines are the equivalent simulations conducted by Schmittner et al. (2013) with the UVic ESM: std (black) and no-bio (purple) on the bottom axis; ki-only (red) and const-gasx (blue) on the top axis. (p. 33)*

[Figure]

*Figure 6: Zonally averaged mean annual surface $\delta^{13}C_{DIC}$ at the end of the sensitivity experiment spin-up simulations (years 9900 to 10,000). The std (black) and no-bio-fract (purple) simulations use the left-hand axis, whilst the ki-fract-only (red) and no-asgx-fract (blue) simulations use the right-hand axis. The dotted lines are the equivalent simulations conducted by Schmittner et al. (2013) with the UVic ESM: std (black) and no-bio (purple) on the left-hand axis; ki-only (red) and const-gasx (blue) on the right-hand axis. (p. 34)*

Figure 3: Why is CaCO₃ in grey? Since there is no fractionation during CaCO₃ formation, then maybe the line from DIC to CaCO₃ should be black. There should be an arrow going from DIC to atm. "Atmosphere" should in fact be "Atm CO₂".

- *CaCO₃ is in grey because the export of CaCO₃ in FAMOUS is represented as an instantaneous redistribution of alkalinity and carbon at depth (i.e. the model doesn't actually carry CaCO₃ as a tracer). This has been clarified in the revised figure caption.*
- *The CaCO₃ formation line was red because, in reality, this process is affected by isotopic fractionation and, in the model, is coded to allow for constant isotopic fractionation. In all of our simulations, we have chosen to set $\alpha_{CaCO3}$ equal to 1.0 because we found that including isotopic fractionation during CaCO₃ formation has a negligible effect on the $\delta^{13}C_{DIC}$ values (as discussed in section 2.2). For clarity, we have revised the figure so that this arrow is orange, and amended the caption to clarify that the orange arrow represents a process that can include a constant isotopic fractionation effect (should future users of the code wish to so), but that this effect has not been included in any of the simulations presented in our manuscript.*
- *We have revised the figure with an arrow going from the DIC pool to the atmospheric pool, but have made it clear in the revised figure caption that the atmosphere doesn't see the outgassed isotopic ratio because atmospheric $\delta^{13}C$ is prescribed.*
- *We have altered "Atmosphere" to "Atmospheric CO₂" to be more accurate.*

*The figure and the corresponding caption have therefore been updated as follows:*

[Figure]

*Figure 2: Schematic overview of the $^{13}C$ implementation in FAMOUS. Blue boxes represent permanent carbon pools. Grey boxes represent temporary carbon pools (note that $CaCO_3$ is a temporary carbon pool because the export of $CaCO_3$ in FAMOUS is represented as an instantaneous redistribution of alkalinity and carbon at depth). The orange box represents the prescribed atmospheric carbon pool. The dashed lines represent fluxes of $^{13}C/^{12}C$. However, note that the outgassed $^{13}C/^{12}C$ has no effect on $\delta^{13}C_{atm}$ because FAMOUS does not currently have a fully interactive carbon cycle. Solid lines represent fluxes of $^{13}C$. Dot-dashed lines represent processes that occur below the lysocline ($\approx$ 2500 m below sea level). The dotted line represents the reflux of detrital material from the seafloor to the surface layer. Red lines represent fractionation effects. The orange line represents isotopic fractionation during calcium carbonate formation ($\alpha_{CaCO3}$), which is included in the code as a user-specified constant. Note that all simulations presented in this study were run without fractionation during calcium carbonate formation (i.e. $\alpha_{CaCO3}$ = 1.0, which is equivalent to a fractionation effect of 0 ‰). (p. 30)*

Figure 7: The authors might consider modifying this figure as it looks like 3 times the same plot. I understand that it might be the point, but maybe best to move some (L95, L97) to SI. The choice of the colorbars could be revised: it is hard (impossible) to see any feature in a), and difficult in the right hand side panels. Are experiments L95 and L97 described in the methods? It would be helpful to also include them in Table 2

- *We have moved the original plot into the supplementary material (now Figure S7) and replaced it in the main manuscript with a 4-panel plot containing the observed values, the std simulation corrected for the mean surface bias, the std simulation, and the difference between the std and observed values:*

[Figure]

***Figure 7:*** *Mean annual surface $\delta^{13}C_{DIC}$ during the 1990s: (a) observations from GLODAPv2 (Key et al., 2015; Olsen et al., 2016), (b) the std simulation corrected for the mean surface bias (0.97 ‰), which is calculated as $\sum$(simulated-observed)/number of observations, (c) the std simulation, and (d) std minus GLODAPv2. (p. 35)*

- *We do not have the same difficulty in seeing the features in the subplots. Perhaps the issue is with the low resolution conversion of the figure in the supplied file. We have therefore attached a PDF of the figure to this response, which is the resolution that will be supplied for the final manuscript. In this version, the colour scale displays the features clearly. We prefer not to change the colour scheme because we selected it very carefully: it is colour-blind friendly and the colour gradations are easy to interpret. We are not aware of an alternative colour scheme that would improve the visualisation of these data.*
- *Yes, experiments L95 and L97 are described in the methods (see p. 8, l. 18 – 23 and p. 10, l. 3 – 7). We have added these simulations to Table 2 for completeness.*

*__Table 2__: Overview of the fractionation factors used in the sensitivity experiments. (p. 28)*

| Simulation | $\alpha_k$ | $\alpha_{aq \leftarrow g}$, $\alpha_{DIC \leftarrow g}$ | $\alpha_p$ |
|---|---|---|---|
| *std* | Standard[1] | Variable[2] | Variable (with $\alpha_{POC \leftarrow aq}$ calculated as per Eq. (10)) |
| *ki–fract-only* | Standard | 1 | 1 |
| *no-asgx-fract* | 1 | 1 | Variable (with $\alpha_{POC \leftarrow aq}$ calculated as per Eq. (10)) |
| *no-bio-fract* | Standard | Variable | 1 |
| *L95* | Standard | Variable | Variable (with $\alpha_{POC \leftarrow aq}$ calculated as per Eq. (11)) |
| *L97* | Standard | Variable | Variable (with $\alpha_{POC \leftarrow aq}$ calculated as per Eq. (12)) |

[1] 0.99919

[2] Calculated as per Eq. (7 – 8)

**Reviewer #2**

General comments
This paper presents the inclusion of stable carbon isotopes, i.e. [13]C, in the FAMOUS model. The authors have evaluated the effect of fractionation by air-sea gas exchange, biology and ocean circulation, and have tested different fractionation parameterisations for biology. This is a very useful development of the model that will be very valuable for paleo studies. This work is well presented and the reasoning is easy to follow. It has already demonstrated its usefulness by showing that the discrepancy between model results and data is likely to be due to biases in the simulated climate as well as the biogeochemical model.

My main concern is that this model should not be used as it is for paleo studies but should be re-tuned, especially the biogeochemical module, as there are very large disagreements between simulated $\delta^{13}C$ and data. This is highlighted by the authors and re-tuning the model is clearly out of the scope of this study, but it might nonetheless be interesting to have a few additional sensitivity simulations to evaluate how much the results could be improved if the biogeochemical model was slightly modified, for example with a modified remineralisation, which could potentially help reduce the model-data disagreements.

*As noted by the reviewer, it is beyond the scope of the current study to retune the model, and as this project is no longer being funded, we are unable to conduct more simulations for inclusion in this manuscript. Nonetheless, we absolutely agree with the reviewer that tuning should be a priority, and have added their suggestion to conduct sensitivity studies with the biogeochemical model (in particular, modifying remineralisation) to section 3.4, where we discuss retuning the model: "In the first instance, further sensitivity studies would provide more insight into the how much our results could be improved by small adjustments to the biogeochemistry in the model (e.g. modifying the remineralisation rate and/or the export ratio)". (p. 15, l.33 – p. 16, l. 2)*

Specific comments
Abstract p.1 l. 7: do you mean "carbon isotopic ratios" instead of isotopic ratios?
Introduction p.2 l.2. The first sentence is almost the same as the first sentence from the abstract: maybe change it?

*In the abstract we were referring to isotopic ratios more widely. For example, $\delta^{18}O$ can be used as a tracer for density, temperature and salinity (Lynch-Stieglitz et al., 1999; Lynch-Stieglitz et al., 1999); $\varepsilon Nd$ can be used as a tracer for water mass provenance (Piotrowski et al., 2004; Rutberg et al., 2000); and $^{231}Pa/^{230}Th$ can be used as a tracer for the rate of overturning and scavenging (Marchal et al., 2000; Henry et al., 2016). However, for clarity, and to avoid repetition in the first sentence of the introduction, we have revised the first sentence of the abstract to "Ocean circulation and the marine carbon cycle can be indirectly inferred from stable and radiogenic carbon isotope ratios ($\delta^{13}C$ and $\Delta^{14}C$, respectively), measured directly in the water column, or recorded in geological archives such as sedimentary micro-fossils and corals." (p. 1, l. 7 – 9)*

p.2. l.2-10. This is not entirely true for [14]C as it also depends on radioactive decay: maybe you could say right at the beginning (after giving the percentages) that C14 is not studied here and only keep [12]C and [13]C in this part.
p2. l.9. You could give the complete definition of $\delta^{13}C$ mode explicitly as this is entirely on $\delta^{13}C$ inclusion it is worth reminding clearly the definition ($\delta^{13}C = ...$).

*In response to the above two comments, we have revised the manuscript to read "There are three naturally occurring carbon isotopes: the stable isotopes $^{12}C$ (98.9 %) and $^{13}C$ (1.1 %), and the radioactive isotope $^{14}C$ ($1.2 \times 10^{-10}$ %), which is also known as radiocarbon (Key, 2001). In this study, we focus on the stable isotopes, with $^{14}C$ being discussed in detail elsewhere (Dentith et al., 2019b), The relative proportions of $^{12}C$*

*and $^{13}C$ in a given oceanic pool (e.g. dissolved inorganic carbon, DIC, or particulate organic carbon, POC) are controlled by ocean circulation and mixing, and mass dependent fractionation during biogeochemical processes such as air-sea gas exchange (Lynch-Stieglitz et al., 1995; Zhang et al., 1995), photosynthesis (e.g. Sackett et al., 1965; Rau et al., 1989; Hollander and McKenzie, 1991; Keller and Morel, 1999), and calcium carbonate formation (Emrich et al., 1970; Turner, 1982; Ziveri et al., 2003). This is typically reported in delta (δ) notation, which is the heavy to light isotope ratio of a sample relative to a standard in per mil (‰) units:*

$$\delta^{13}C = \left( \frac{^{13}C/_{12}C_{sample}}{^{13}C/_{12}C_{standard}} - 1 \right) \times 1000. \qquad (1)$$

*Oceanic $\delta^{13}C$ is primarily used to track individual water masses (Curry and Oppo, 2005), study past changes in the carbon cycle (e.g. de la Fuente et al., 2017), and investigate changes in ocean circulation on glacial-interglacial timescales (e.g. Spero and Lea, 2002; Campos et al., 2017). It has also been used to constrain air-sea gas exchange rates (Gruber and Keeling, 2001) and to estimate the uptake of anthropogenic carbon by the global oceans (Quay et al., 1992, 2003)." (p. 2, l. 2 – 15)*

p.3 l. 5-8. You should also include the LOVECLIM model. What about Genie?
*Our intention was to provide illustrative examples of $^{13}C$-enabled models across a range of complexities as opposed to a complete list of all $^{13}C$-enabled models, but we have added these additional examples to Table 1 (p. 26 – 27).*

p.3 l.19. There is an arrow that should be deleted between "studying" and "complex" .
*This typographical error has been corrected (p. 3, l. 20).*

p5. Line 16. Sea ice does not change salinity: this is out of the scope of the study but probably needs to be modified in the model. . .
*Acknowledged. We note that an iceberg meltwater is included, however, and have amended the text as follows: "...do not affect salinity distributions (an area for future development), although the model does include an iceberg meltwater flux (Smith et al., 2008)." (p. 5, l. 15 – 16)*

P5. Line 27-28. This seems at odds with what is said later. From what I understand from this paper both the physical model AND the biogeochemical model are responsible for carbon isotopes mismatch between simulation results and data and disentangling between the two is not done here.
*We can see the possible confusion/contradiction, and have revised the manuscript as follows:*

- *Section 2.1 (p. 5, l. 25 – 28): "Previous studies have found that errors in biogeochemical simulations are largely driven by biases in the physical ocean circulation (i.e. inaccuracies in the climate or ocean model to which the ecosystem model has been coupled; Doney, 1999; Doney et al., 2004; Najjar et al., 2007). Thus, simulating carbon isotopes in a more complex ecosystem model would not necessarily yield substantially better results."*
- *Section 3.4 (p. 15, l. 15 – 19): "In contrast with earlier studies, we have demonstrated that the new carbon isotope scheme in FAMOUS is sensitive to biases in both physical and biogeochemical processes. The simulated $\delta^{13}C_{DIC}$ distributions reflect known physical inaccuracies (such as over-deep NADW and weak convection in the sub-polar North Pacific Ocean) and have allowed us to identify previously undisclosed biogeochemical biases (e.g. in the representation of remineralisation). The new tracer therefore offers excellent potential as a holistic tuning target for recalibrating FAMOUS in the future."*

Results and discussion

p. 10 l. 15 / Figure 4. I would start with standard results before looking at the sensitivity experiments to be able to compare these sensitivity experiments with the standard one. So, on Figure 4 I would add the standard simulated $\delta^{13}C$ first as (a) and then the other 3 sensitivity simulations as b-d, which would also be more coherent with having the 4 simulations on Figure 5.

*We have chosen to present the results of the sensitivity experiments before the standard (std) results because this is, first and foremost, a model development study and we want to highlight that our new $^{13}C$ tracer is responding as expected to the physical processes (which have known biases) and the biogeochemical processes (where we have identified new biases). We think that beginning with the std results immediately raises questions regarding how they compare to observations, and consequently, why the simulated values are higher than observed. This would detract from the validation of the new isotope scheme. Therefore, we prefer not to restructure the results or figures as suggested.*

p.10 line 19. Is this a simulation that you actually did to verify this or just discussion? Please specify.

*We conducted this simulation to verify that our code is correct (i.e. that no carbon is being created or destroyed), but we did not include the results in our manuscript because they matched what we expected to see ($\delta^{13}C_{DIC}$ equilibrating at -6.5 ‰) and so are not very interesting. For clarity, we have revised this sentence to "If there is no fractionation during either air-sea gas exchange or photosynthesis, the ocean equilibrates at a uniform value of -6.5 ‰, in line with the atmosphere (simulation not shown)." (p. 10, l. 13)*

p.11 line 8. Could you quickly remind the reader what this simulation is (to avoid looking for it earlier in the text)?

*We have revised this sentence to "When only biological fractionation effects are included (no-asgx-fract), $\delta^{13}C_{DIC}$ values in the surface ocean range between -7.65 ‰ in the eastern equatorial Pacific and -3.89 ‰ in the eastern equatorial Atlantic (Figure 4c), representing a shift of -1.15 ‰ to +2.61 ‰ relative to no isotopic fractionation." (p. 11, l. 3 – 5)*

p.14 l.18. Could you test your hypothesis for the cause of the model-data mismatch due to the export ratio and remineralisation rate vs biases in ocean circulation by running additional sensitivity experiments? Testing the ocean circulation is probably more difficult, but modifying the export ratio and/or remineralisation to evaluate if this could have a large contribution to the mismatch is probably easier.

*There is a lot of scope for future tuning and exploration with this model set up, and we are excited to see this work carried forward. However, this effort is beyond the scope of this illustrative study. We have, however, added these suggestions to section 3.4: "
[revised manuscript text omitted]